# Automated ECG Arrhythmia Classification Using Feature Images with Common Matrix Approach-Based Classifier

**DOI:** 10.3390/s25041220

**Published:** 2025-02-17

**Authors:** Ali Kirkbas, Aydin Kizilkaya

**Affiliations:** Department of Electrical and Electronics Engineering, Faculty of Engineering, Pamukkale University, Denizli 20160, Türkiye; akizilkaya@pau.edu.tr

**Keywords:** arrhythmia classification, common matrix approach (CMA), electrocardiogram (ECG), Fourier decomposition method (FDM), time–frequency (T-F) analysis

## Abstract

This paper seeks to solve the classification problem of cardiac arrhythmias by using a small number of electrocardiogram (ECG) recordings. To offer a reasonable solution to this problem, a technique that combines a common matrix approach (CMA)-based classifier model with the Fourier decomposition method (FDM) is proposed. The FDM is responsible for generating time–frequency (T-F) representations of ECG recordings. The classification process is performed with feature images applied as input to the classifier model. The feature images are obtained after two-dimensional principal component analysis (2DPCA) of data matrices related to ECG recordings. Each data matrix is created by concatenating the ECG record itself, the Fourier transform, and the T-F representation on a single matrix. To verify the efficacy of the proposed method, various experiments are conducted with the MIT-BIH, Chapman, and PTB-XL databases. In the assessments using the MIT-BIH database under the inter-patient paradigm, we achieved a mean overall accuracy rate of 99.81%. The proposed method outperforms the majority of recent efforts, yielding rates exceeding 99% on nearly five performance metrics for the recognition of V- and S-class arrhythmias. It is found that, in the classification of four types of arrhythmias using ECG recordings from the Chapman database, our model surpasses recent works by reaching mean overall accuracy rates of 99.76% and 99.45% for the raw and de-noised ECG recordings, respectively. Similarly, five different forms of arrhythmias from the PTB-XL database were recognized with a mean overall accuracy of 98.71%.

## 1. Introduction

Heart rhythm disorders, also known as cardiac arrhythmias, are abnormalities or irregularities in the heartbeat. These disorders can be mild and harmless, but in some cases they can be acute and even lethal. Rapid and accurate detection of these symptoms is essential for taking medical precautions. The electrocardiogram (ECG), which captures the heart’s electrical activity from a human body surface, is one of the most common and noninvasive diagnostic tools used for monitoring cardiac disorders [1].

Analysis and assessment of recorded or instantaneously collected ECG signals often require expert knowledge. However, cardiac arrhythmias are typically the result of long-term effects. Thus, an event-by-event examination by cardiac specialists to determine the type and existence of arrhythmia is time-consuming and may lead to an inaccurate diagnosis. Consequently, automatic arrhythmia classification from ECGs helps in the diagnosis and treatment of cardiac complaints in patients and facilitates medical interventions [2]. Classification is the final stage of ECG analysis. Prior to this, ECG signals are typically subjected to preprocessing and feature extraction operations [3]. Noise filtration and segmentation are generally applied to ECG heartbeat signals in the preprocessing step. Subsequently, feature extraction is applied to the ECG waveforms or their transformed versions to capture discriminative and subtle characteristics describing the arrhythmias in the ECG. The classification process is carried out by leveraging the time-, frequency-, or time–frequency-based features of the ECG [4]. It has been declared that features based on time–frequency (T-F) representations of ECGs better reflect dynamic changes and discriminative information and provide high accuracy in determining arrhythmia types when used with classifiers [5]. Motivated by these findings, in this paper, we focus on automatic arrhythmia classification using T-F representations of ECG signals. The T-F representation provides a 2D map of how the energy of a signal is distributed throughout the T-F plane. In this way, it is possible to observe when and at which frequencies the signal properties change. Given the non-stationary nature of ECG signals, T-F analysis is well suited for identifying subtle and discriminative features describing the arrhythmias within them. This makes the use of T-F representations of ECGs appealing for arrhythmia classification. In the past few years, numerous studies combining various learning model topologies have been put forth in this direction.

For ECG classification, the authors of [6,7,8,9] propose a variety of 2D convolutional neural network (CNN) architectures that use T-F representations obtained from the short-time Fourier transform (STFT) of heartbeats as inputs. The other study group suggests using CNN models fed with continuous wavelet transform (CWT) scalogram representations of heartbeat signals [10,11,12,13,14]. In a recent paper [15], a new method has been proposed that combines scalograms and phaseograms obtained from CWT with a simple CNN architecture, yielding an improvement in arrhythmia classification performance. In addition, the T-F images coming from the modified frequency slice wavelet transform of ECG signals have been used together with deep neural networks [16] and CNNs [17] for arrhythmia classification. Allam et al. [18] advanced an ECG beat classification system for five classes of arrhythmia. This system involves the use of T-F images obtained from the Stockwell transform (ST) along with a 2D residual network. In another study of ECG classification, a 2D CNN model in conjunction with a bi-directional long short-term memory network for the automated detection of atrial fibrillation (AF) pathology is developed [19]. This deep learning model uses the T-F images from the chirplet transform (CLT) of ECG signals as input. Yet another ECG classification approach for AF detection is a deep CNN model that processes the mean of Kalman-based spectro-temporal estimations of ECG beat segments [20]. The authors of [21] offer a ventricular arrhythmia detection method that feeds machine learning-based classifiers with T-F images resulting from the pseudo-Wigner–Ville distribution (PWVD). A CNN model for classifying five types of arrhythmia is presented in a recent study [22]. This model contains multi-head self-attention mechanisms and employs time-reassigned synchro squeezing transforms of ECGs as input. Several studies are introduced to evaluate the classification performance of CNN models fed with different T-F representations. In [23], three kinds of T-F analysis methods (STFT, CWT, and PWVD) are evaluated on the fixed 2D CNN to discriminate twelve ECG rhythm classes. Similarly, the STFT and stationary wavelet transform outputs of ECG beats have been fed into two different CNN models for the detection of AF episodes [24].

As opposed to the aforementioned studies, other researchers have chosen to use transfer learning for the identification and categorization of arrhythmias instead of building deep learning networks from the ground up. Cinar and Tuncer [25] propose using a hybrid structure consisting of an AlexNet model and support vector machine (SVM) for the classification of ECG signals, including abnormal arrhythmia, normal sinus rhythm, and congestive heart failure. To this end, data features are extracted from the STFT spectrograms given to the AlexNet input and are then classified by the SVM algorithm. In a similar study [26], a DenseNet-SVM structure is used to process the spectrogram images of the ECG data for the classification of four arrhythmia types. Similarly, Toma et al. [27] have presented a comparative study on the efficiency of six pre-trained CNN-based classifiers for the diagnosis of fifteen types of cardiac arrhythmias using the STFT of long-duration ECG data. Another framework based on pre-trained CNNs using the T-F spectrograms of ECG records for detecting and classifying cardiac arrhythmias has been proposed in a recent paper by Tripathi et al. [28]. This paper leverages the superlet transform (SLT) to extract 2D T-F spectrograms from ECG recordings of three cardiac conditions, such as ventricular fibrillation, AF, and a healthy heart. Eltrass et al. [29] have proposed a method for feeding the pre-trained AlexNet CNN model by using T-F images obtained from the constant-Q non-stationary Gabor transform of ECG data. This method aimed to classify three types of heart disorders, including congestive heart failure, arrhythmia, and normal sinus rhythm. Additionally, Zhang et al. [30] have devised a heartbeat classification method that combines a hybrid T-F analysis technique with transfer learning based on the ResNet-101 model. In this method, a combination of the Hilbert transform and WVD has been used for the T-F analysis of 1D ECG recordings. The resulting 2D T-F maps are then submitted to the learning model for classification into a total of nineteen categories of heartbeats. In [31,32], VGGNet-based pre-trained CNN models coupled with CWT-based T-F map inputs have been proposed for ECG heartbeat classification. Alqudah et al. [33] provided a comparative study on the performance of four pre-trained CNN architectures in the classification of six distinct types of ECG arrhythmias. These deep learning models use four T-F representations as input, which are bi-spectrum, third-order cumulant, log-scale STFT, and Mel-scale STFT.

It is clear from all of these studies that T-F representations combined with deep learning architectures have achieved satisfactory performance in ECG rhythm classification without the need for handcrafted feature extraction. While these studies are touted as automatic approaches or systems for ECG classification, they are problem-based in the sense that they deal with the design of deep learning architectures as well as the acquisition of T-F representations. Both aspects need user-defined parameters. For example, the STFT, WT, and PWVD techniques used for the T-F representations of ECG signals are dependent on the selection and application of window functions [23]. Similarly, the CLT and ST require the selection of Gaussian window parameters [19,34]. Also, the SLT requires determining some design parameters, such as the number of wavelet functions, cycle numbers, and standard deviation value [28]. As with the T-F analysis methods, the design of deep learning models that use T-F images of ECG data for classification tasks is also problem-based and user-defined. The design of deep learning architectures relies heavily on determining parameters such as hidden layers, nodes per layer, learning rate, batch size, epochs, regularization parameters, and momentum parameters.

Based on the above considerations, it becomes clear that a new method should be adopted for ECG arrhythmia classification in the context of T-F representations integrated with intelligent learning models. This paper aims to provide an automatic end-to-end ECG classification method that is as independent of user-based decisions to derive T-F representations as possible, does not require the development of deep learning architectures, and offers high performance for the classification of ECG arrhythmias. To this end, a technique combining the Fourier decomposition method (FDM) with the common matrix approach (CMA), called FDM-CMA, is proposed. The FDM is a data-driven adaptive signal decomposition tool that can be applied to nonlinear and non-stationary time series and has been used to provide T-F representations of ECG signals. It is free of parameter settings and has a precise mathematical basis. This method decomposes a given signal into a finite number of band-limited orthogonal components, termed analytic Fourier intrinsic band functions (AFIBFs) [35]. Unlike the STFT and CWT, the FDM provides high-resolution T-F representations reflecting the characteristics of the considered signal through the AFIBFs [35,36,37,38]. The T-F representation of the signal is constructed using the instantaneous amplitudes and instantaneous phases of these complex-valued AFIBFs. Each one of the AFIBs represents a different frequency band and contains distinctive information that characterizes the signal. Taking this fact into consideration, the FDM has been applied to the ECG arrhythmia classification task. The ECG signals representing each arrhythmia class have unique characteristics and are defined by different numbers of distinct AFIBFs. The T-F representations created using these AFIBFs are decisive for the definition of arrhythmia classes. The ability to identify a signal via AFIBFs makes the FDM attractive for classification applications, such as in recognizing hand movements [39] and in detecting epileptic seizures [40], alcoholism [41], myocardial infarction [42], sleep apnea [43,44], biometric identity [45], and hypertension [46]. In implementing the FDM, we have taken advantage of its recently developed fast version [47]. In the proposed method, the classification process is performed using feature images obtained after two-dimensional principal component analysis (2D PCA) [48] of data matrices related to ECG recordings. Each data matrix is created by concatenating the ECG signal itself, the Fourier transform, and the T-F representation on a single matrix. The feature matrices are then fed into a CMA-based classifier model for the detection of arrhythmias. The CMA is an image classification method that was initially created by Turhal et al. [49] for face recognition applications. This classifier model does not require user-defined parameters and performs effectively with small amounts of training data. During the training phase, it obtains a common matrix defining the shared characteristics of feature images in each unique class. The testing step leverages the common matrices for classification. We adopt the CMA based on the Gram–Schmidt orthogonalization (GSO) technique, following the instructions given in [50].

The main contributions and novelties of this study are outlined as follows:A CMA-based classifier model fused with the FDM is proposed for ECG arrhythmia classification.The FDM, a fully adaptive data-driven signal decomposition method, is used to obtain the T-F representation of the ECG signal.The classification process is performed using feature matrices. The feature matrices are obtained after 2D PCA of data matrices related to ECG recordings. Each data matrix is created by concatenating the ECG record itself, the Fourier transform, and the T-F representation on a single matrix.A CMA-based model is used as a classifier model, which can operate independently of the user parameter settings and performs effectively with a small amount of training data.The proposed FDM-CMA method shows superior performance in arrhythmia classification experiments performed on the MIT-BIH, Chapman, and PTB-XL databases using a small amount of training data for each arrhythmia class.To the authors’ knowledge, this study is the first attempt to use FDM-based T-F representations of ECG signals together with a CMA-based classifier for ECG arrhythmia classification.

The remainder of this paper is organized as follows: Section 2 presents the proposed technique for classifying ECG arrhythmias, along with the mathematical basis of the FDM as a T-F representation tool, the CMA as a classifier model, and 2D PCA as a feature matrix generator. Dataset descriptions and evaluation criteria used for the validation of the proposed method are given in Section 3. The presentation of the experimental results and performance comparisons with up-to-date methods are covered in Section 4. Section 5 offers key conclusions and recommendations for further research.

## 2. Proposed Method

This study proposes an automated method based on the FDM and CMA to classify arrhythmias, leveraging T-F representations of ECG signals. The FDM is used to generate T-F representations of ECGs, while the CMA is utilized as a classifier model. Accordingly, the proposed method is built upon two primary pillars—feature matrix generation and classification. The general layout of the proposed method is shown in Figure 1.

### 2.1. Feature Matrix (Image) Generation

Feature matrices, or feature images, are used in this work as inputs to the CMA-based classifier model. The feature images of ECG signals are obtained by applying the 2D PCA to data matrices. Each data matrix is created by concatenating the ECG signal itself, the Fourier transform, and the T-F representation on a single matrix.

#### 2.1.1. An Overview of the Fourier Decomposition Method

In this study, the T-F representations of ECG signals are obtained using the Fourier decomposition method (FDM), which is a fully adaptive data-driven method. The FDM decomposes a signal into a finite number of band-limited orthogonal components, i.e., AFIBFs. The sum of the Fourier intrinsic band functions (FIBFs), which are the real parts of AFIBFs, provides the full description of the signal [35]. As detailed in the Section 1, the ability to describe a signal through FIBFs has made the utilization of the FDM in various classification studies attractive [39,40,41,42,43,44,45,46]. In these studies, a predetermined quantity of features—such as kurtosis, entropy, Lp-norms, energy, etc.—are computed from every FIBF and fed into machine learning-based classifiers. It is evident from these studies that the FDM results in superior performance in identifying hand movements [39], epileptic seizures [40], alcoholism [41], myocardial infarction [42], sleep apnea [44], biometric identity [45], and hypertension [46]. But the specific features that are obtained from FIBFs based on the user’s choices could result in subpar performance and vary depending on the applications, as observed in [43]. In light of this, we concentrate on T-F representations that encompass all information associated with the signal’s AFIBFs instead of certain characteristics that are computed from FIBFs.

Before applying the FDM to an N-point ECG signal, the following normalization is performed on the signal:(1)$$s\left(n\right)=\frac{\left[\bar{s}\left(n\right)-\mu \right]}{max\left\{\bar{s}\left(0\right),\bar{s}\left(1\right),\;\ldots ,\bar{s}\left(N-1\right)\right\}},$$
for 0≤n≤N−1. In Equation (1), $\bar{s}\left(n\right)$ is the N-point raw ECG signal, s(n) is the N-point ECG signal after normalization, and μ is the mean value of the N-point raw ECG signal. Normalization allows for the study of signals from various databases by eliminating the scale effects of the data while maintaining the distribution characteristics.

The FDM starts with calculating the discrete Fourier transform (DFT) coefficients, $S(k)=(1/N)\sum_{n=0}^{N-1}s(n)e^{-j2\pi kn/N}$ for k=0, 1,…,N−1. The N-point inverse DFT (IDFT) of ${\left\{S\left(k\right)\right\}}_{k=0}^{N-1}$ turns into the original signal:(2)$$s\left(n\right)=\sum_{k=0}^{N-1}S\left(k\right)e^{j2\pi kn/N}$$

Equation (2) can be expressed equivalently for even and odd values of N. The mathematical basis of the FDM for even values of N can be found in [47]. To avoid repetition, we concentrate on presenting the mathematics of the FDM for odd-length signals. Accordingly, the equivalent form of (2) is given as follows for an odd value of N,(3)$$s\left(n\right)=S\left(0\right)+Re\left\{z\left(n\right)\right\},$$
where Re{z(n)} is the real part of the analytical signal z(n) that is defined by(4)$$z\left(n\right)=2\sum_{l=1}^{(N-1)/2}S(l)e^{j2\pi ln/N}.$$

In the framework of the FDM, the analytical signal defined by (4) can be decomposed into a finite number of band-limited orthogonal components as(5)$$z\left(n\right)=\sum_{m=1}^{M}w_{m}\left(n\right),$$
where $w_{m}(n)=a_{m}(n)e^{j\phi_{m}(n)}$ stands for the mth AFIBF, $a_{m}(n)$ stands for its instantaneous amplitude (IA), $\phi_{m}(n)$ stands for its instantaneous phase (IP), and M stands for the total number of AFIBFs that automatically emerge from the signal under analysis.

Low-to-high and high-to-low (HTL) frequency scan procedures are used to decompose (4) into (5), yielding two different sets of AFIBFs [35]. The HTL frequency scanning procedure was adopted to analyze ECG signals. Each AFIBF is determined so that its IP is a monotonically increasing function,(6)$$\varphi_{m}\left(n\right)=\phi_{m}\left(n+1\right)-\phi_{m}\left(n\right)\geq 0,$$
where $\varphi_{m}(n)$ expresses the IP differentiation for the mth AFIBF. Using $\varphi_{m}(n)$, an estimate of the IF of the mth AFIBF is obtained by $f_{m}\left(n\right)=\varphi_{m}\left(n\right)/2\pi$, for n=0, 1,…,N−1.

Within the framework of the HTL frequency scanning procedure, the AFIBFs of the analytical signal $z\left(n\right)$ are obtained as follows [35]:(7)$$w_{m}(n)=2\sum_{k=N_{m}}^{N_{m-1}-1}S(k)e^{j2\pi kn/N}=a_{m}(n)e^{j\phi_{m}(n)},$$
for m=1,2,…,M−1 with $N_{0}=(N+1)/2$ and $N_{M}=1$.

In order to define the AFIBFs using (7), the frequency index ${\left\{N_{m}\right\}}_{m=1}^{M-1}$ is determined. This is an iterative process, in which the mth frequency index $N_{m}$ is searched in the range $\left[N_{m-1}-1,N_{M}\right]$ for m=1,2,…,M−1 and requires a total of $N_{m-1}-N_{M}$ iterations [47]. For the mth AFIBF, this searching process is performed over(8)$$w_{m}^{\left(p\right)}\left(n\right)=2\sum_{k=N_{m-1}-p}^{N_{m-1}-1}S\left(k\right)e^{j2\pi kn/N}=a_{m}^{\left(p\right)}\left(n\right)e^{j\phi_{m}^{\left(p\right)}(n)},$$
for $p=1,2,\ldots ,N_{m-1}-N_{M}$, and n=0,1,…,N−1.

Equation (8) represents the IDFT in the pth iteration, and $a_{m}^{\left(p\right)}\left(n\right)$ and $\phi_{m}^{\left(p\right)}(n)$ express the corresponding IA and IP, respectively. As introduced in [47], the IDFT coefficients in each iteration can be calculated in a computationally efficient manner using the N-point inverse fast Fourier transform (IFFT) algorithm:(9)$$v_{m}^{\left(p\right)}=\left[v_{m}^{\left(p\right)}\left(0\right),\;v_{m}^{\left(p\right)}\left(1\right),\ldots ,v_{m}^{\left(p\right)}\left(N-1\right)\right]=ifft\left(u_{m}^{\left(p\right)},N\right),$$
where(10)$$u_{m}^{\left(p\right)}=\left[0_{1\times \left(N_{m-1}-p\right)},\;2S\left(N_{m-1}-p\right),\ldots ,\right.\left.2S\left(N_{m-1}-N_{M}\right),\;0_{1\times \left(N-N_{m-1}\right)}\right]$$

The IPs of $v_{m}^{\left(p\right)}$, represented by $\phi_{m}^{\left(p\right)}\left(n\right)$ for n=0, 1,…,N−1, are assessed after each iteration to see if they satisfy the requirement in (6). As shown in [47], meeting the criterion defined by(11)$$\epsilon_{m}^{\left(p\right)}=-Re\left\{t_{m}^{\left(p\right)}\right\}⊙Im\left\{v_{m}^{\left(p\right)}\right\}+Im\left\{t_{m}^{\left(p\right)}\right\}⊙Re\left\{v_{m}^{\left(p\right)}\right\}\geq 0$$
will be enough to ensure the positivity of the IPs of $v_{m}^{\left(p\right)}$. Here, ⊙ symbolizes the element-wise multiplication, $Im\left\{.\right\}$ states the imaginary part, and $t_{m}^{\left(p\right)}$ is an array constructed from the entries of $v_{m}^{\left(p\right)}$ found by (9), i.e., $t_{m}^{\left(p\right)}=\left[v_{m}^{\left(p\right)}\left(1\right),\;v_{m}^{\left(p\right)}\left(2\right),\ldots ,v_{m}^{\left(p\right)}\left(N-1\right),v_{m}^{\left(p\right)}\left(0\right)\right]$.

The p values satisfying the criterion in (11) are collected on a set of(12)$$\Omega_{m}=\left\{p\in \beta_{m}\;:\epsilon_{m}^{\left(p\right)}\geq 0\right\},$$
with $\beta_{m}=\left\{1,\;2,\ldots ,N_{m-1}{-N}_{M}\right\}$ for m=1, 2, …, M−1. Following that, the frequency index relating to the mth AFIBF of (7) is then obtained as(13)$$N_{m}=N_{m-1}-\max\left(\Omega_{m}\right),$$
where $\max\left(\Omega_{m}\right)$ indicates the maximum value in the set of $\Omega_{m}$. The frequency indices obtained from (13) are substituted into (7). Finally, the T-F representation of the ECG signal is produced using the IFs, $f_{m}\left(n\right)$, and the IAs, $a_{m}(n)$, of all the AFIBFs.

#### 2.1.2. Data Matrix Generation

In this study, the arrhythmia classification task is carried out by feeding the CMA-based classifier model with feature images. The feature images representing ECG signals are obtained by leveraging data matrices. As demonstrated by ablation experiments in the subsequent sections, the design of a data matrix plays a critical role in classifying arrhythmias with high accuracy. In the proposed method, the data matrix for an ECG signal was established using the ECG signal itself, its Fourier transform, and its T-F representation, as follows:(14)$$D=\left[\begin{array}{ccc} - & S_{T-F} & -\\ - & S_{F} & -\\ - & s & - \end{array}\right]$$
where $s=\left[s\left(0\right),s\left(1\right),\;\ldots ,s\left(N-1\right)\right]$ is the vector of samples of the normalized ECG signal and $S_{F}=\left[\tilde{S}(0),\tilde{S}(1),\;\ldots ,\tilde{S}(N-1)\right]$ is the vector of DFT coefficients of the normalized ECG signal. The entries of vector $S_{F}$ are in the form of $\tilde{S}\left(k\right)=\left|S(k)\right|/max\left\{\left|S(0)\right|,\;\left|S(1)\right|,\ldots ,\left|S(N-1)\right|\right\}$, where $\left|.\right|$ is the absolute operator. Additionally, $S_{T-F}$ indicates the clipped and normalized version of a T-F representation matrix that is obtained by applying the FDM to the normalized ECG signal. By using clipping, we restrict our T-F representation to the 0–60 Hz frequency range because there is no information above that range. In the sequel, the T-F representation was normalized to its maximum value. As a result, in the arrhythmia classification task, we worked with data matrices of size 62 × *N*.

#### 2.1.3. Feature Image Extraction

Since the IA values for frequencies outside the IFs are taken as zero during the creation of the T-F map, the discernibility of ECG signal classes decreases. To overcome this drawback, data matrices representing each ECG signal class are subjected to 2D PCA [48]. This process results in feature images whose size is smaller than the data matrices defined in (14).

In the 2D PCA framework, a unique projection matrix is obtained that transforms the data matrices of ECG signals in relation to each arrhythmia class into feature matrices. Let us suppose that $\left\{D_{1},\;D_{2},\;...,\;D_{R}\right\}$ is the set of R data matrices representing any arrhythmia class. The in-class covariance (scatter) matrix of this data matrix set is evaluated by(15)$$\Phi =\frac{1}{R}\sum_{k=1}^{R}{\left(D_{k}-D_{a}\right)}^{T}\left(D_{k}-D_{a}\right),$$
where $D_{a}$ is the average data matrix [48].

The covariance matrix calculated using (15) is symmetric, and thus can be represented by $\Phi =Q\Lambda Q^{T}$, where Q is an *N* × *N* orthonormal matrix with orthonormal eigenvectors $\left\{q_{1},\;q_{2},\;\ldots ,\;q_{N}\right\}$ in its columns, and **Λ** is an *N* × *N* diagonal matrix with eigenvalues $\left\{\lambda_{1},\;\lambda_{2},\;\ldots ,\;\lambda_{N}\right\}$ in its main diagonal. After ordering the eigenvalues in a descending manner, the corresponding eigenvectors of the covariance matrix Φ are obtained, which is denoted by $\left\{p_{1},\;p_{2},\;\ldots ,\;p_{N}\right\}$. The first *d* eigenvectors are employed to form the projection matrix expressed by(16)$$P=\left[\begin{array}{cccc} | & | & \ldots & |\\ p_{1} & p_{2} & \ldots & p_{d}\\ | & | & \ldots & | \end{array}\right].$$
which is unique for each ECG arrhythmia class. Let P be the projection matrix for a specific arrhythmia class and D be the data matrix of an ECG signal in that class. Accordingly, the feature matrix corresponding to this ECG signal is generated by(17)F=DP.

It is worthwhile to note that the data matrices used in this work have dimensions of 62 × *N*. Therefore, the projection matrices of arrhythmia classes and the feature matrices of ECG signals in those classes have dimensions of *N* × *d* and 62 × *d*, respectively. The number of eigenvectors corresponding to the *d* largest eigenvalues of the covariance matrix defined in (15) may be determined by taking into account the criterion described in [50]. In this study, classification experiments were carried out by selecting d = 20 and d = 30. Therefore, the effect of different *d* values on the classification performance was also observed. It should be noted that the training phase of the classifier model produces as many projection matrices as the number of arrhythmia classes. These projection matrices are used in the testing phase of the classifier model to produce feature images of the ECG signals. A schematic representation of the feature matrix generation process is plotted in Figure 2.

### 2.2. CMA-Based Classifier Model

The CMA is a subspace-based classification method developed to perform recognition directly on image data. It has found applications in face recognition [49,50] and edge detection [51]. This approach is predicated on the idea that each feature matrix within a pattern class is made up of common and difference matrices.

The common matrix describes the common properties of feature matrices in a specific class, whereas the difference matrix includes the variable qualities of each feature matrix. Thus, a feature matrix of a specific class can be expressed as(18)$$X_{j}^{\left(i\right)}=X_{com}^{\left(i\right)}+X_{j,dif}^{\left(i\right)},$$
where $X_{com}^{\left(i\right)}$ and $X_{j,dif}^{\left(i\right)}$ are the common and difference matrices of a feature matrix $X_{j}^{\left(i\right)}$, respectively.

While the difference matrix is distinct for every feature matrix in a given class, the common matrix is unique for all feature matrices in that class [50]. Hence, there is no need to calculate the difference matrices of all of the feature matrices in a class. Computing the difference matrix for a single feature matrix will suffice to find the common matrix of a relevant class. The difference matrix is calculated using a set of orthonormal basis matrices spanning a difference subspace for each class. Finding the common matrices and the corresponding orthonormal basis matrices for all of the classes establishes the training portion of the CMA-based classifier. The testing portion of this classifier leverages these matrices to determine the class to which a particular ECG signal belongs.

#### 2.2.1. Training Phase

Let $L_{i}$ be the total number of feature matrices in the ith class. The feature matrices corresponding to the data matrices of ECG signals in the ith class are represented by a set of $X_{1}^{\left(i\right)}$, $X_{2}^{\left(i\right)}$, …, $X_{L_{i}}^{\left(i\right)}$ matrices, each of which has a size h×d. The training phase of the CMA starts with determining the difference subspace of each individual class. To this end, any feature matrix in the ith class is selected as a reference. Here, for the sake of computational simplicity, $X_{1}^{\left(i\right)}$ is chosen as a reference matrix. Accordingly, the *within-class difference matrices* defined by(19)$$Y_{r}^{\left(i\right)}=X_{r+1}^{\left(i\right)}-X_{1}^{\left(i\right)}$$
are found for i=1,2,…,K and $r=1,2,\ldots ,L_{i}-1$, where K is the total number of arrhythmia classes.

In order to determine the orthonormal bases of the difference subspace for each individual class, the within-class difference matrices defined in (19) are first orthogonalized using the GSO technique, as follows [50]:$$Z_{1}^{\left(i\right)}=Y_{1}^{\left(i\right)},$$(20)$$Z_{k}^{\left(i\right)}=Y_{k}^{\left(i\right)}-\sum_{l=1}^{k-1}\left(\langle Y_{k}^{\left(i\right)},Z_{l}^{\left(i\right)}\rangle /{\left\|Z_{l}^{\left(i\right)}\right\|}_{F}^{2}\right)Z_{l}^{\left(i\right)},$$
for $k=2,\;3,\ldots \;,\;L_{i}-1$, where $\langle Y_{k}^{\left(i\right)},Z_{l}^{\left(i\right)}\rangle =tr\left(Y_{k}^{\left(i\right)T}Z_{l}^{\left(i\right)}\right)$ stands for the inner product of $Y_{k}^{\left(i\right)}$ and $Z_{l}^{\left(i\right)}$ and is equal to the sum of elements on the main diagonal of $Y_{k}^{\left(i\right)T}Z_{l}^{\left(i\right)}$. Also, ${\left\|{Z_{l}}^{\left(i\right)}\right\|}_{F}$ implies the Frobenious norm of $Z_{l}^{\left(i\right)}$ and is equal to the positive value of the square root of $\langle Z_{l}^{\left(i\right)},Z_{l}^{\left(i\right)}\rangle$.

The orthogonal matrices resulting from (20) are divided to their Frobenius norms to obtain orthonormal variants thereof:(21)$$Q_{r}^{\left(i\right)}=Z_{r}^{\left(i\right)}/{\left\|Z_{r}^{\left(i\right)}\right\|}_{F},\;r=1,2,\ldots ,L_{i}-1.$$

The set of $Q_{1}^{\left(i\right)}$, $Q_{2}^{\left(i\right)}$, …, $Q_{L_{i}-1}^{\left(i\right)}$ matrices establishes the orthonormal bases of difference subspace $D^{\left(i\right)}$ for class i and spans this subspace as well. The difference matrix $X_{j,dif}^{\left(i\right)}$ is then calculated using (21), as follows,(22)$$X_{j,dif}^{\left(i\right)}=\sum_{r=1}^{L_{i}-1}\langle X_{j}^{\left(i\right)},Q_{r}^{\left(i\right)}\rangle Q_{r}^{\left(i\right)},$$
which corresponds to the projection of any feature matrix $X_{j}^{\left(i\right)}$ onto the difference subspace $D^{\left(i\right)}$. Finally, the common matrix of class i is obtained by putting (22) into (18):(23)$$X_{com}^{\left(i\right)}=X_{j}^{\left(i\right)}-X_{j,dif}^{\left(i\right)}.\;$$

Keep in mind that each class has its own common matrix. It will therefore be sufficient to compute (22) and (23) for a single feature matrix relating to each class.

The training phase of the CMA is completed after obtaining the set of orthonormal basis matrices, $\left\{{Q_{1}}^{\left(i\right)},{Q_{2}}^{\left(i\right)},\ldots ,{Q_{L_{i}-1}}^{\left(i\right)}\right\}$, and the common matrix, $X_{com}^{\left(i\right)}$, for all of the arrhythmia classes under consideration. A schematic representation of the training process for class i is given in Figure 3.

#### 2.2.2. Testing Phase

Let T be an h×d-sized feature matrix corresponding to the unknown ECG signal. To determine the arrhythmia class to which this test signal belongs, T is first projected onto the difference subspace of class i, resulting in the difference matrix of T.(24)$$T_{dif}^{\left(i\right)}=\sum_{r=1}^{L_{i}-1}\langle T,Q_{r}^{\left(i\right)}\rangle Q_{r}^{\left(i\right)}.\;$$

According to (23), the corresponding common matrix is obtained by(25)$$T_{com}^{\left(i\right)}=T-T_{dif}^{\left(i\right)}.\;$$

The classification procedure is completed by assigning the test matrix T to the ith arrhythmia class in a way that satisfies the minimal Frobenius norm determined by(26)$$W=\underset{1\leq i\leq K}{min}\left\{W^{\left(i\right)}\right\},\;$$
where$$W^{\left(i\right)}={\left\|T_{com}^{\left(i\right)}-X_{com}^{\left(i\right)}\right\|}_{F}^{2}\;$$
expresses the Frobenius norm between the common matrices attained by (23) and (25). Figure 4 presents a schematic of the testing phase for a given ECG signal.

## 3. Dataset Descriptions and Evaluation Metrics

The arrhythmia classification performance of the proposed method is evaluated based on five performance metrics using three public databases, such as the MIT-BIH, Chapman, and PTB-XL.

### 3.1. The MIT-BIH Arrhythmia Database

The MIT-BIH arrhythmia database [52] consists of 48 dual-channel ECG recordings collected from 47 individuals, 25 males aged 32 to 89 years and 22 females aged 23 to 89 years. Each record is about 30 min long and digitized at 360 Hz. Of the recordings used for the selection of arrhythmia classes, 23 contain common arrhythmias, while the remaining contain rare yet clinically important arrhythmias. In this database, R-peak locations are also provided for each heartbeat. As shown in Table 1, all records in this database are defined under sixteen heartbeat types and merged into five classes based on the AAMI standard [52]. These arrhythmia classes are defined as non-ectopic beats (N), supraventricular ectopic beats (S), ventricular ectopic beats (V), fusion beats (F), and unknown beats (Q). In our experiments, we employ 44 records from the lead II channel, while 4 records—102, 104, 107, and 217—are excluded due to their inadequate quality. Q-class heartbeats cannot be meaningfully classified due to the small number of records. Consequently, Q-class heartbeats are not included in performance evaluations in our experiments. The non-ectopic class in Table 1 is also expanded to three classes: normal beats (N), left bundle branch block beats (L), and right bundle branch block beats (R). Thus, performance evaluations are made considering six arrhythmia classes: N, L, R, V, S, and F.

### 3.2. The Chapman Database

The Chapman database [53] consists of twelve-lead ten-second ECG recordings collected from 10,646 patients with 500 samples per second that feature 11 common rhythms and 67 additional cardiovascular conditions, all of which are labeled by professional cardiologists. The patients include 5956 males and 4690 females, 17% of whom suffer from normal sinus rhythm and 83% from at least one abnormality. These disorders were recorded in the age groups 51–60, 61–70, and 71–80, representing 19.82%, 24.38%, and 16.9%, respectively. In accordance with cardiologists’ suggestions, some rare types of arrhythmias in this database are hierarchically merged into four upper-level arrhythmia types [53], as shown in Table 1. The publisher of this database has also supplied the de-noised counterparts of the raw ECG recordings that are tainted by baseline wandering effects, power-line interference, and electromyographic and random noise. Since the proposed FDM-CMA method uses a single-lead ECG signal, raw and de-noised ECG recordings from the lead II channel are used in our experiments.

### 3.3. The PTB-XL Database

The PTB-XL is a large, publicly available database consisting of 21,837 clinical 12-lead ECG recordings collected from 18,885 patients, 52% male and 48% female, with ages ranging from 0 to 95 [54]. Each record is 10 s long, sampled at 100 Hz and 500 Hz. All records contain labels for 44 cardiac disorders in diagnostic subclasses that have been combined into five major arrhythmia classes, namely ST/T change (STTC), hypertrophy (HYP), conduction disturbance (CD), normal ECG (NORM), and myocardial infarction (MI). In this study, experimental analysis is conducted on a total of 16,244 ECG records that do not contain multi-labels; 9069, 1708, 535, 2532, and 2400 of these records belong to the NORM, CD, HYP, MI, and STTC classes, respectively. Since the proposed FDM-CMA technique is based on single-lead ECG signals, PTB-XL recordings belonging to lead I with a 500 Hz sampling rate are used in our experiments.

### 3.4. Metrics for Performance Evaluation

We invoke five evaluation criteria, namely sensitivity (Sen), specificity (Spe), positive predictive rate (Ppr), F1-score, and overall accuracy (OA), to quantitatively evaluate the validity of the proposed method. The overall accuracy is adopted as the metric against which the probability of an individual being correctly classified by a test is measured. When calculating these metrics, the confusion matrix is taken into account.

## 4. Results and Discussion

This section covers a number of experiments conducted to assess the arrhythmia classification performance of the proposed FDM-CMA method. ECG records extracted from the MIT-BIH, Chapman, and PTB-XL databases are used in these experiments. For this assignment, a PC running Matlab with an Intel Core i9-9900K CPU running at 3.60 GHz and with 32 GB of RAM is used.

### 4.1. Performance Evaluations Using the MIT-BIH Database

Performance evaluations based on the MIT-BIH database are carried out over segmented ECG signals. The R-peak locations provided by the MIT-BIH database are utilized to segment ECG signals, as stated in [55] (p. 9). This is achieved by taking 150 sample points before and after the R-peak, resulting in fixed-length ECG segments of 301 sample points. The feature matrices of the ECG segments are produced by following the procedures described in Section 2.1.1, Section 2.1.2 and Section 2.1.3. To assess their influence on the performance of arrhythmia classification, two feature matrices of sizes 62 × 20 and 62 × 30 were generated with *d* = 20 and *d* = 30, respectively, for every ECG segment.

The inter-patient paradigm is adopted in the experiments conducted on the MIT-BIH database. According to this paradigm, the training set, which contains 50,957 beats, and the test set, which contains 49,470 beats, are composed of heartbeats coming from different individuals [56]. Given the inter-patient paradigm, we create our training set by randomly selecting 1000 heartbeats from every arrhythmia class. Since the S and F classes have less than 1000 beats, all of them are used for training. As a result, Table 2 shows that our classifier model is trained using 5358 heartbeats and evaluated using 49,470 heartbeats.

The performance of the proposed arrhythmia classification technique on the MIT-BIH dataset described in Table 2 was evaluated through 100 independent runs which correspond to Monte Carlo cross-validation (MCCV). Although the heartbeats for the test and training sets come from different patients under the inter-patient paradigm, the classifier model was still trained independently using 100 random training sets and evaluated on the test set. Therefore, there is no overfitting problem for the proposed classification technique. After every random run, the confusion matrix for the classification results is used to compute evaluation metrics. The evaluation metrics derived from 100 random runs are provided in Table 3 with the mean and standard deviation (*std*) values. This table indicates that the mean values generally closely resemble the single-run metrics. The low *std* values suggest that the proposed classifier provides stable predictions on the test set and thus can be trusted in terms of its single-run results.

The Sen and F1-score metrics clearly reveal that misclassifications are concentrated in class F. This is because the classifier model was trained with less F class data than the others. When the feature matrix was enlarged to 62 × 30, the recognition of F-class beats improved; a 3% rise in F1-score and an approximate 5% increase in sensitivity are the indicators of this improvement. The findings in the table demonstrate that beats in all classes except class F may be classified nearly flawlessly using the proposed technique in both feature matrix dimensions (62 × 20 and 62 × 30); the mean Sen and Spe values above 99% are an indication of this. The overall accuracy rates also support these inferences. With 62 × 20 feature matrices, the proposed technique achieves a mean overall accuracy rate of 99.73% with a standard deviation of 1.01 × 10⁻^3^, while when 62 × 30 feature matrices are used, the mean overall accuracy rate achieved is 99.81% with a standard deviation of 5.56 × 10^−4^.

Although Table 3 presents a comprehensive performance analysis, the prediction results for any of the 100 random runs are shown in Figure 5 to give insight to readers. This figure shows that, employing 62 × 30 feature matrices, the proposed technique produces 36, 1, and 58 misclassifications for classes N, L, and F, respectively. Fortunately, all of the heartbeats relating to the R, V, and S classes are detected correctly. When 62 × 20-sized feature matrices are used, the classifier yields 8, 8, 1, 2, 1, and 87 misclassifications for the N, L, R, V, S, and F classes, respectively. As is seen in Figure 5, the impact of enlarging the size of the feature matrix to 62 × 30 is high for class F. The Sen and F1-score metrics given in Table 3 verify this fact. However, the results in Table 3 and Figure 5 demonstrate that the classification of beats belonging to arrhythmia classes other than class F is not significantly impacted by expanding the feature matrix size to 62 × 30. A similar situation applies to the overall accuracy rates of 99.78% and 99.84%, which are calculated from the confusion matrices in Figure 5a,b, respectively.

The time consumed for classifying six arrhythmias from the MIT-BIH database is given in a class-wise manner in Figure 6 for two different feature image sizes. These timings also include the time required for feature extraction. This figure shows the mean values of time consumption obtained from 100 random trials for each arrhythmia class. According to the single-run results in Figure 6, the average test time per patient utilizing the proposed technique with 62 × 20 (62 × 30) feature images is around 22 (24) ms.

Finally, the performance of the proposed method is compared with the most recent ECG classification algorithms [57,58,59,60,61,62,63,64,65,66,67], all of which use the MIT-BIH database under the inter-patient paradigm. Table 4 shows that the proposed method gives the highest overall accuracy among the compared ones. The detection of V- and S-class arrhythmias is important because of their role in the diagnosis of arrhythmias that cause sudden cardiac arrest. At that point, the proposed method exhibits the best statistics for the V and S classes across all evaluation criteria. This observation is further supported by the prediction results given in Figure 5.

### 4.2. Performance Evaluations Using the Chapman Database

The raw and de-noised ECGs from the Chapman database are utilized for performance evaluation in an effort to verify the generalization capacity of the proposed method in detecting ECG arrhythmias. The inter-patient classification paradigm is naturally adopted in the experiments since all of the records in this database are from different individuals.

Unlike the experiments conducted on the MIT-BIH database, the segmentation process is omitted; the feature matrix generation process mentioned in Figure 2 is executed directly on 5000-sample ECG records of the SB, GSVT, AFIB, and SR classes. Using the Chapman dataset detailed in Table 1, the effectiveness of the proposed technique for classifying these arrhythmia classes was assessed over 10 random MCCV simulations. The feature matrices of the ECG recordings are produced by following the procedures outlined in Section 2.1.1, Section 2.1.2 and Section 2.1.3. As in previous experiments, the classification processes are carried out with 62 × 20 and 62 × 30 feature matrices for every ECG recording.

#### 4.2.1. Experiments on Raw (Noisy) ECG Recordings

In this experiment, the efficacy of the proposed FDM-CMA technique is validated by 10 independent MCCV simulations conducted on the raw ECG recordings fetched from the Chapman database. In that spirit, the classifier model is trained with 1000 randomly selected recordings from each arrhythmia class reported in Table 1, with the remaining ECG recordings being reserved for testing. The classification results obtained based on 62 × 20 and 62 × 30 feature images for single and multiple runs are presented in Table 5. The table findings show that the proposed technique achieves better classification performance with 62 × 20 feature images than with 62 × 30 ones. As a result of applying the 2D PCA on data matrices relating to noisy ECG records, the 62 × 20-sized feature matrices have less noise than the 62 × 30 ones. The better performance provided by the 62 × 20 feature images can be explained by the increased discernibility resulting from the reduced effect of noise. The mean metrics originating from 10 random runs of the proposed technique are compatible with the single-run metrics, as is seen in Table 5. Also, the proposed classifier appears to be reliable in terms of its single-run results due to its low standard deviation values, which indicate that it produces consistent predictions on the test set. The proposed technique exhibits superior classification performance with 62 × 20 feature images, reaching rates above 99.5% across all metrics. Additionally, the mean overall accuracy rates are calculated as 99.76% and 95.98% for the feature images of 62 × 20 and 62 × 30, respectively. The test results for any of the 10 random trials are shown in Figure 7 to give insight to readers.

As evidenced by the results in Figure 7, the proposed method attains high recognition performance with feature matrices of 62 × 20. For all classes, only 12 of 6646 ECG records are classified incorrectly. On the other hand, the proposed method leads to 263 misclassifications with 62 × 30 feature images. According to these findings, the proposed method yields a 99.82% (96.04%) overall accuracy rate for feature images of 62 × 20 (62 × 30). These results are also consistent with the performance metrics given in Table 5. From this table, the average values for the Sen, Spe, Ppr, and F1-score metrics are calculated as 99.76% (94.99%), 99.95% (98.79%), 99.76% (95.54%), and 99.76% (94.89%), respectively.

#### 4.2.2. Experiments on De-Noised ECG Recordings

In this experiment, the performance of the proposed technique is validated by means of 10 independent MCCVs implemented on the de-noised ECG recordings collected from the Chapman database. In each implementation, the classifier model is trained with 1000 recordings chosen at random from each of the arrhythmia classes reported in Table 1, with the remaining ECG recordings being reserved for testing, as in the previous experiment. It should be pointed out that 40 records of the SVT subclass are not included in the test set because they contain just zeros [68]. Thus, the trained model is evaluated on 6606 ECG records. The classification results obtained based on 62 × 20 and 62 × 30 feature images for single and multiple runs of the proposed FDM-CMA technique are presented in Table 6. The confusion matrices of the predictions obtained from 10 random trials were used to calculate the performance metrics, which are represented by the mean and standard deviation values in parentheses in this table.

In the classification of four types of arrhythmias from the de-noised ECG recordings, the proposed technique yields higher scores using 62 × 30 feature images, as shown in Table 6. In particular, Sen and F1-score values for all arrhythmia classes clearly show the effect of employing 62 × 30 feature images in classification. Table 6 shows that, for the feature images of 62 × 20, the GSVT class has the lowest average Spe and Ppr metrics across all classes, which means that almost all of the incorrectly classified records from other classes were allocated to the GVST class. Enlarging the feature matrix to 62 × 30 caused a significant increase in the Spe and Ppr metrics of the GSVT class, while it caused a small decrease in the Spe and Ppr metrics of the AFIB class. This means that almost all of the incorrectly classified records from other classes were allocated to the AFIB class. The prediction outcomes in Figure 8 for any of the ten random trials corroborate these assessments.

It is evident from Figure 8 that, supporting the evaluations made for multiple runs, better performance is achieved with 62 × 30 feature matrices in de-noised ECG recordings; only 25 of 6606 ECG records are classified incorrectly. On the other hand, a total of 1989 misclassifications across all arrhythmia classes are obtained with feature images of 62 × 20. Accordingly, the proposed technique yields overall accuracy rates of 69.89% (70.10%) and 99.62% (99.45%) for feature matrices of 62 × 20 and 62 × 30, respectively, where the values in brackets state the mean overall accuracy rates resulting from the ten MCCV simulations. Table 6 also includes the single-run metrics derived from the confusion matrices in Figure 8. Based on 62 × 20 (62 × 30) feature images, the suggested method achieves average rates of 78.79% (99.57%), 90.69% (99.88%), 83.80% (99.51%), and 74.99% (99.54%) for the Sen, Spe, Ppr, and F1-score metrics, respectively, in the classification of four types of arrhythmias. Following the application of the 2D PCA to data matrices associated with de-noised ECG recordings, the 62 × 30-sized feature matrices bear more distinctive information than the 62 × 20 ones. Consequently, more information representing the arrhythmia class may account for the improved performance provided with 62 × 30-sized feature matrices compared to 62 × 20-sized ones.

The time consumed for classifying four types of arrhythmias pertaining to raw and de-noised ECG recordings from the Chapman database is given in a class-wise manner in Figure 9 for two different feature image sizes. These times also include the time required for feature extraction. This figure shows the mean values of time consumption caused by ten random trials for each arrhythmia class. With 62 × 20 (62 × 30) feature images, the proposed technique diagnoses a given raw ECG record in about 1.91 (1.79) seconds, as shown in Figure 9a. According to Figure 9b, these timings are measured as 0.38 (0.39) seconds for the diagnosis of a de-noised record. These results clearly show that, even when the format, quantity, and technique are the same for both types of ECG recordings, training and testing the suggested approach with de-noised ECG records takes less time than with noisy ones. Compared to the de-noised ECG records, noisy records contain a greater number of frequency components. Therefore, the frequency scanning process in the FDM takes a long time to decompose every noisy record into its AFIBF.

A class-wise performance comparison with the up-to-date ECG classification methods [68,69,70] is introduced in Table 7, in which all of the methods compared use ECG records from the Chapman database. In [68], a deep neural network model fusing 1D CNN and LSTM networks is developed to detect rhythm classes from each of the 12-lead ECG signals. As reported in this work, lead II signals provide the best results, with an overall accuracy of 96.13. Lee and Shin [69] propose a framework that utilizes beat score map images to train a 2D CNN for arrhythmia classification. Their model achieves an overall accuracy of 88.83% in detecting four classes of arrhythmias. Most recently, Shi et al. [70] offered a self-supervised learning model that blends contrastive and generative learning schemes, demonstrating an overall accuracy of 95.11% in the classification of four types of arrhythmia using 12 leads of ECG records from the Chapman database. The proposed technique surpasses the others in terms of all metrics and provides mean overall accuracy rates of 99.76% and 99.45% in classifying arrhythmia types related to raw and de-noised ECG records, respectively. It is valuable that this classification performance is achieved by training the proposed classifier model with less data and then testing it with more data.

### 4.3. Performance Evaluations Using the PTB-XL Database

To verify the success of the proposed technique in classifying different types of arrhythmia, ECG recordings retrieved from the PTB-XL database were used. As in the experiments conducted on the Chapman database, feature image generation was performed directly on 5000-sample ECG records, and the performance of the proposed technique was validated by means of 10 independent MCCVs. In each run, recordings that belong to the five classes of the PTB-XL database were chosen at random. Accordingly, 4450 recordings—1000 samples from each of the classes NORM, CD, MI, and STTC, plus 450 samples from class HYP—were employed for training, and the remaining 11,794 records were for testing. The classification results obtained based on 62 × 20 and 62 × 30 feature images for single and multiple runs of the proposed technique are shown in Table 8. The values in brackets represent the means and standard deviations corresponding to the metrics calculated from the confusion matrices of the predictions resulting from each run.

The metric values in Table 8 show that the classifier model fed with 62 × 20 feature images provides better predictions. Such a result can be explained by the fact that the records provided from the PTB-XL database are noisy. After 2D PCA, the discernibility between classes increased because the noise effect was reduced in the 62 × 20-sized feature images compared to in the 62 × 30 ones. The fact that the mean Sen value for the HYP class is close to zero is an indication that this arrhythmia class cannot be recognized. Also, the Ppr and F1-score are obtained as NaN due to zero false positives. For classes other than HYP, sensitivity levels above 97% and F1-score values above 93% indicate that most records in these classes are accurately classified. These evaluations are supported by the test results shown in Figure 10, which correspond to any one of ten random trials.

As shown in Figure 10, the FDM-CMA technique gives high classification performance with 62 × 20 feature images; only 175 of 11,794 ECG records are classified incorrectly. On the other hand, using 62 × 30 feature images results in 2730 misclassifications. Records belonging to the HYP class are not identified in both feature image sizes, as was previously stated. This is due to the small number of HYP records in the PTB-XL database, as encountered in [70]. According to the confusion matrices in Figure 10, the proposed technique yields overall accuracy rates of 98.52% and 76.26% based on the feature images of 62 × 20 and 62 × 30, respectively. These single-run values are compatible with the mean overall accuracy rates of 98.71% and 76.77%, which emerge from ten MCCVs based on 62 × 20 and 62 × 30 feature images, respectively.

The time consumed for classifying five types of arrhythmia relating to the ECG recordings from the PTB-XL database is given in a class-wise manner in Figure 11. These times also include the time required for feature extraction. This figure shows the mean values of time consumption obtained from ten random trials for each arrhythmia class. With 62 × 20 (62 × 30) feature images, the proposed technique diagnoses a given ECG record in about 1.31 (1.29) s.

Additionally, the proposed technique appears to perform better than the newly released approach [70]. In detecting five arrhythmia classes from the PTB-XL database, the proposed technique achieves an overall accuracy rate of 98.71%, whereas the method of [70] does so at 77.06%. Since the records in this database are noisy, the feature images obtained as a result of dimension reduction (2D PCA) with a size of 62 × 20 contain less noise than those with a size of 62 × 30. Consequently, the reduced influence of noise may account for the improved performance provided with 62 × 20-sized feature images compared to 62 × 30-sized ones.

### 4.4. General Comparison with Other Recent Studies

In this section, we compare the FDM-CMA method with up-to-date arrhythmia classification methods that use the MIT-BIH, Chapman, and PTB-XL databases. Table 9 shows the comparative results based on the overall accuracy, average Sen, and average F1-score for each method.

Table 9 shows that the suggested FDM-CMA technique outperforms the others in predicting various arrhythmia classes in the MIT-BIH and Chapman databases, with an average F1-score above 98%, a Sen value above 97%, and overall accuracy performance close to 100%. Similarly, it is observed that the proposed method achieves about a 15% higher Sen percentage and a 21% higher OA percentage for the PTB-XL database compared to Shi et al. [70]. Since at least one of the Ppr metrics for the HYP class was obtained as indeterminate (NaN) in 10 random trials, the F1-score percentage was found to be NaN.

### 4.5. Ablation Experiments

The efficacy of the CMA-based classifier model in classifying arrhythmias can be directly impacted by the data matrix from which the feature image (matrix) is generated for the training process in the proposed FDM-CMA method. In the experiments conducted, each data matrix was created using the ECG signal itself, the Fourier transform, and the T-F representation, as defined in (14). After applying the 2D PCA to the data matrices, feature images of 62 × 20 and 62 × 30 were generated for use in the training and testing processes.

Ablation experiments were conducted to examine the impact of various data matrix configurations on the classification performance of the proposed FDM-CMA technique. To that end, three additional data matrix forms were established, as follows:$$D_{1}=\left[\begin{array}{ccc} - & S_{T-F} & - \end{array}\right]$$$$D_{2}=\left[\begin{array}{ccc} - & S_{T-F} & -\\ - & s & - \end{array}\right]$$$$D_{3}=\left[\begin{array}{ccc} - & S_{T-F} & -\\ - & S_{F} & - \end{array}\right]$$
where the entries of these three data matrices are as in data matrix D that is defined in (14). In the framework of the ablation study, feature images of 60 × *d* and 60 × *d* for $D_{1}$, 61 × *d* and 61 × *d* for $D_{2}$ and $D_{3}$, and 62 × *d* and 62 × *d* for D were produced when the 2D PCA was applied to the $D_{1}$, $D_{2}$, $D_{3}$, and D data matrices created using the same records. In the simulations, *d* was chosen to be 10, 20, 30, and 40. In the sequel, these feature images were employed in the training and testing stages of the FDM-CMA technique. These data matrices and, consequently, the feature matrix configurations were assessed for their effects on the classification performance of the proposed FDM-CMA technique. Table 10 shows the performance comparisons in terms of the sensitivity (Sen), F1-score, and overall accuracy (OA) metrics.

Table 10 shows that the best classification statistics in all three databases are provided by the feature matrices obtained from data matrix configuration **D**. At this point, feature images with the dimension parameter of *d* = 20 provide satisfactory performance, especially for noisy data records from the MIT-BIH, Chapman, and PTB-XL databases. However, feature images with the dimension parameter smaller than 30 result in poor classification performance for de-noised data records from the Chapman database.

## 5. Conclusions

Accurately diagnosing arrhythmias in their earliest stages is crucial for identifying heartbeat-related ailments. For this purpose, a CMA-based classifier model combined with the FDM, called FDM-CMA, is proposed. The validity and efficacy of the proposed method in ECG arrhythmia classification are assessed on three publicly accessible databases: MIT-BIH, Chapman, and PTB-XL. In experiments conducted to detect six classes of arrhythmia, N, L, R, V, S, and F, using the MIT-BIH database, we achieved the highest mean overall accuracy of 99.81 under the inter-patient classification paradigm. Achieving rates of 99% across four evaluation metrics highlights the effectiveness of the proposed method in detecting V- and S-class arrhythmias. This implies that arrhythmias that can result in sudden cardiac arrest can be diagnosed using the proposed method.

Additionally, it was observed that the proposed FDM-CMA method shows satisfactory performance in diagnosing four types of arrhythmia, termed SB, GSVT, AFIB, and SR, associated with the raw and de-noised ECG records collected from the Chapman database. The highest mean overall accuracy rates of 99.76% and 99.45% were reached for the noisy and de-noised ECG recordings, respectively. Similarly, the proposed method presents favorable results in detecting five types of arrhythmia—NORM, CD, HYP, MI, and STTC—using the ECG records from the PTB-XL database, resulting in the highest overall accuracy rate of 98.71%. The empirical results of the performance metrics clarify that the proposed technique exhibits a performance that surpasses existing methodologies documented in the literature.

Achieving high overall accuracy rates with the ECG records belonging to the different databases provides insight into the robustness and generalizability of the proposed method for classifying different types of arrhythmias. Additionally, when compared with the classification results of state-of-the-art ECG classification methods using the same databases, it is observed that the proposed method performs better with a small amount of training data. Therefore, the FDM-CMA method is thought to be a potential replacement for existing methods in the classification of arrhythmias since it avoids application-based user parameters as much as possible. It is also evident that the intermediate outputs of this method may have the potential to serve as input for deep learning algorithms and/or different classifier structures. The ablation experiments demonstrate the effect of the data matrix formation used for the training and testing stages of the CMA-based classifier model.

Despite the impressive results, there are some limitations in applying the proposed method to ECG signals. One of the main limitations of the FDM-CMA method is the imbalanced distribution of samples across classes in the training set. It requires ECG beat labels for classification. The computational complexity of obtaining the T-F representations based on the FDM is dependent on the signal being analyzed. The FDM spends a long time analyzing distorted signals, which restricts its practical applicability. This limitation can be overcome by exposing the signal to some preprocessing operations.

In the future, our research will have two essential goals for practical application: employing tensor algebra to implement the CMA-based classifier and handling the processing load of the proposed FDM-CMA technique on cloud servers.

## Figures and Tables

**Figure 1 sensors-25-01220-f001:**
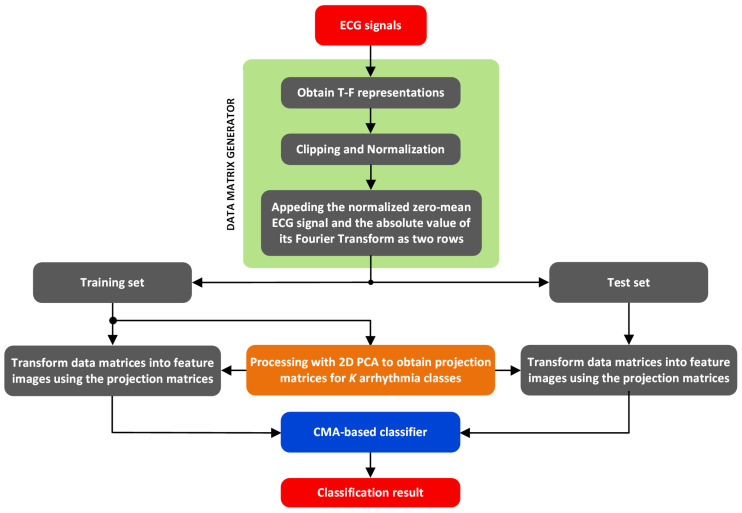
The overall structure of the proposed method for the ECG arrhythmia classification.

**Figure 2 sensors-25-01220-f002:**
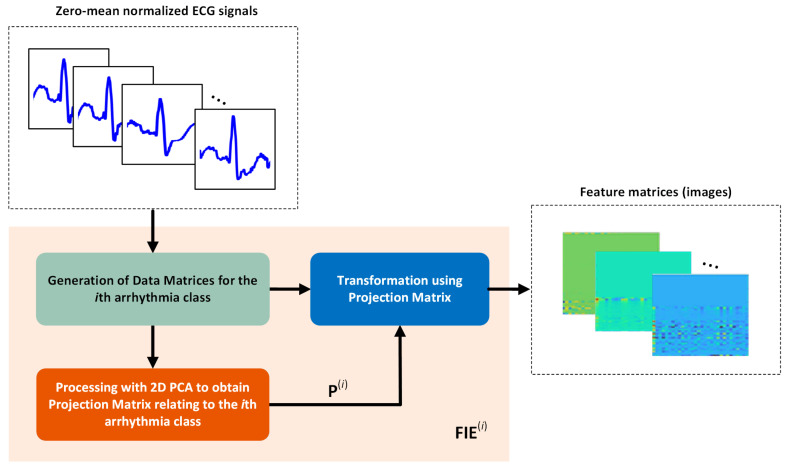
Feature image (matrix) extraction process for *i*th arrhythmia class.

**Figure 3 sensors-25-01220-f003:**
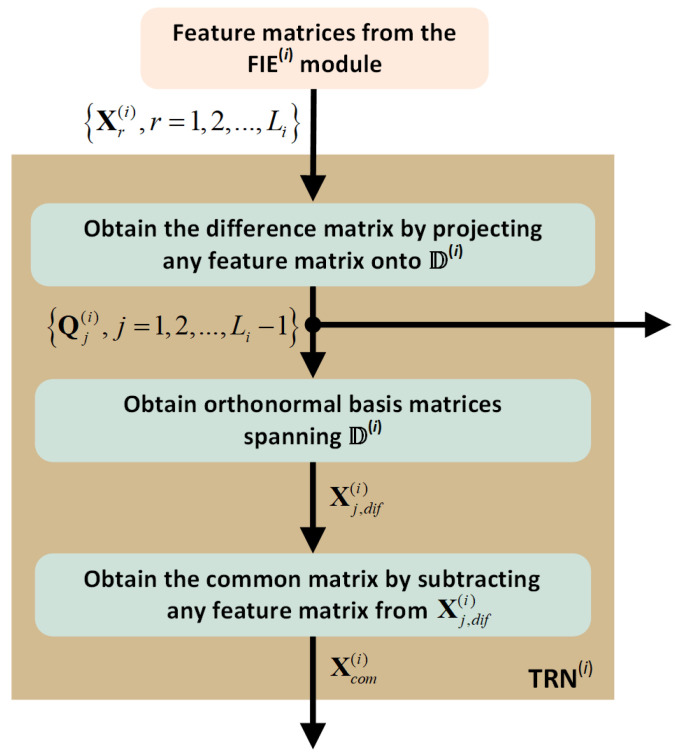
Training module for i th arrhythmia class.

**Figure 4 sensors-25-01220-f004:**
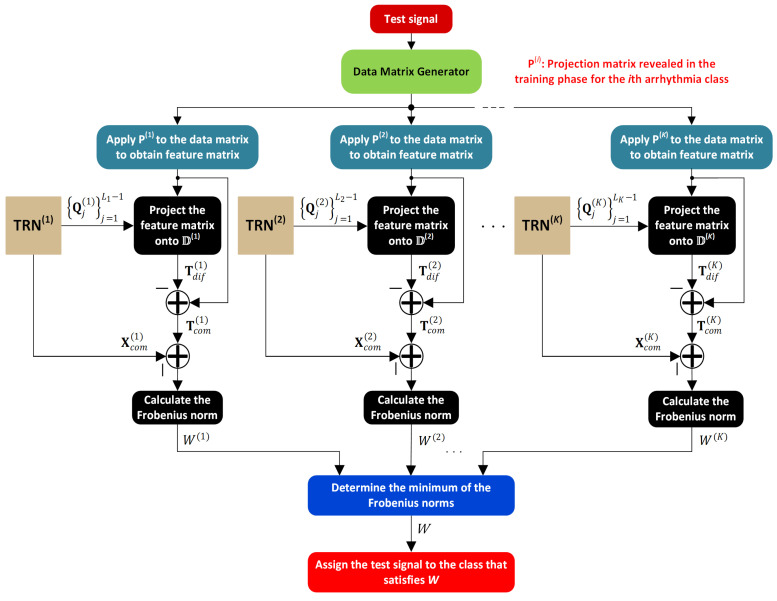
Test module for K arrhythmia classes.

**Figure 5 sensors-25-01220-f005:**
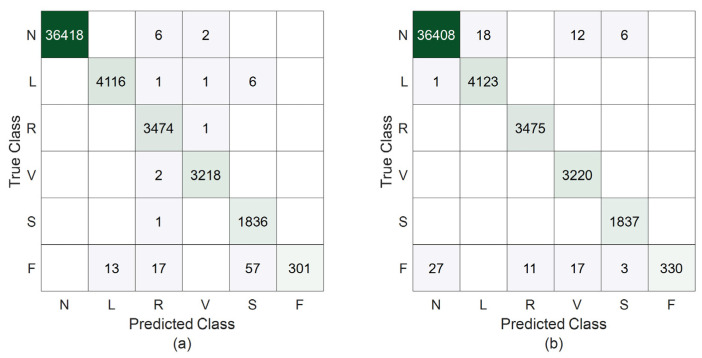
The confusion matrices resulting from training and testing the proposed technique using the heartbeats from the MIT-BIH database. (**a**) The results for 62 × 20 feature matrices; (**b**) the results for 62 × 30 feature matrices.

**Figure 6 sensors-25-01220-f006:**
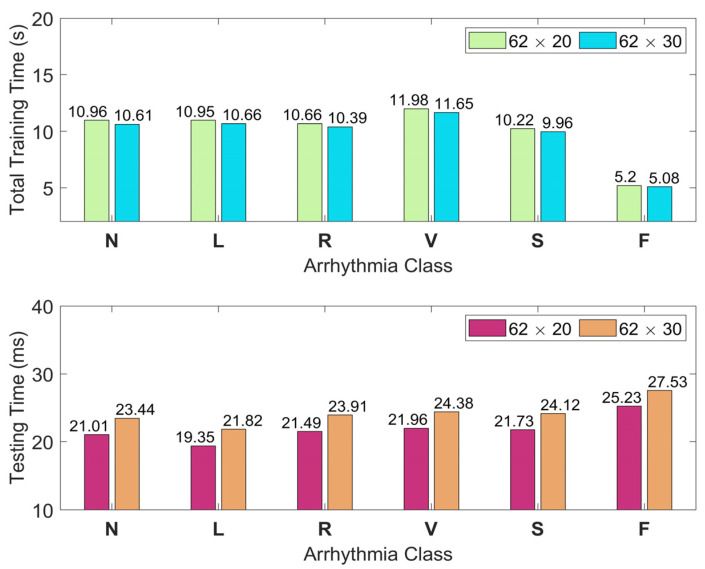
The training and testing times in a class-wise manner for the classification of six arrhythmias from the MIT-BIH database. Here, testing times are per patient.

**Figure 7 sensors-25-01220-f007:**
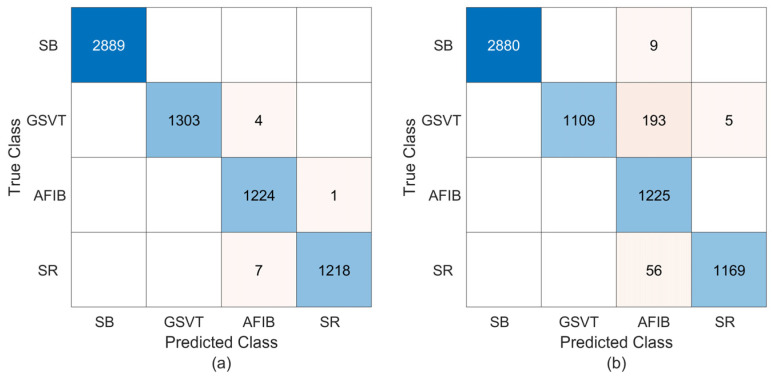
The confusion matrices coming from training and testing the proposed technique using the raw ECG recordings of the Chapman database. (**a**) The results for 62 × 20 feature matrices; (**b**) the results for 62 × 30 feature matrices.

**Figure 8 sensors-25-01220-f008:**
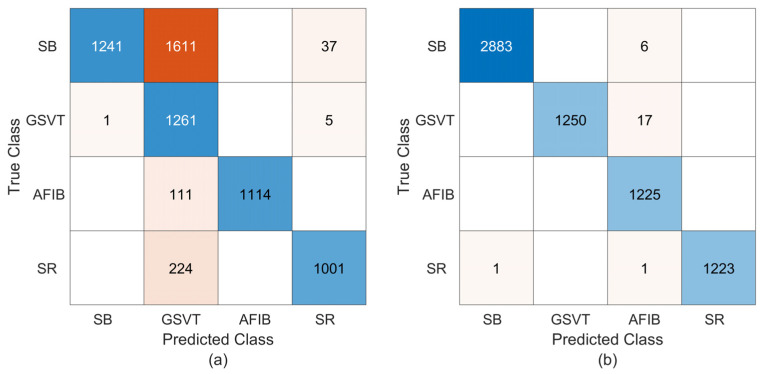
The confusion matrices coming from training and testing the proposed technique using the de-noised ECG recordings of the Chapman database. (**a**) The results for 62 × 20 feature matrices; (**b**) the results for 62 × 30 feature matrices.

**Figure 9 sensors-25-01220-f009:**
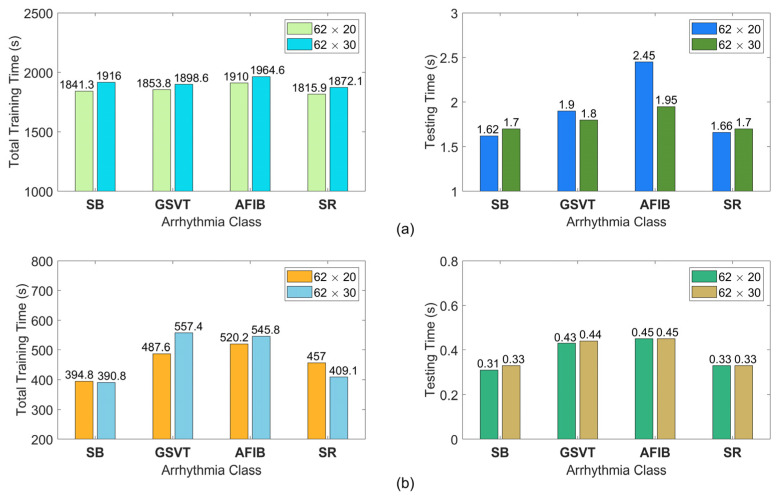
The training and testing times for the classification of four types of arrhythmia relating to raw and de-noised ECG records from the Chapman database. Here, testing times are per patient. (**a**) The results for raw ECG records; (**b**) the results for de-noised ECG records.

**Figure 10 sensors-25-01220-f010:**
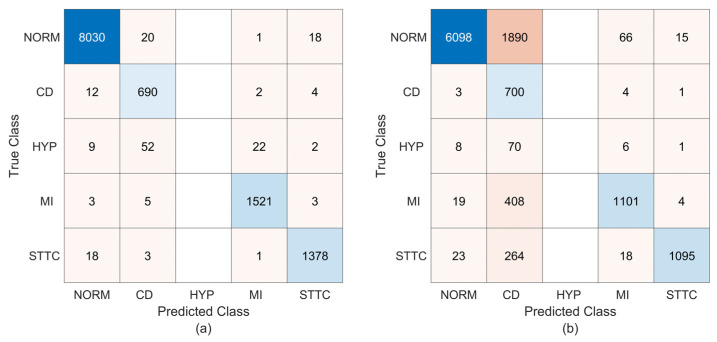
The confusion matrices obtained from training and testing the proposed technique using the ECG recordings of the PTB-XL database. (**a**) The results for 62 × 20 feature matrices; (**b**) the results for 62 × 30 feature matrices.

**Figure 11 sensors-25-01220-f011:**
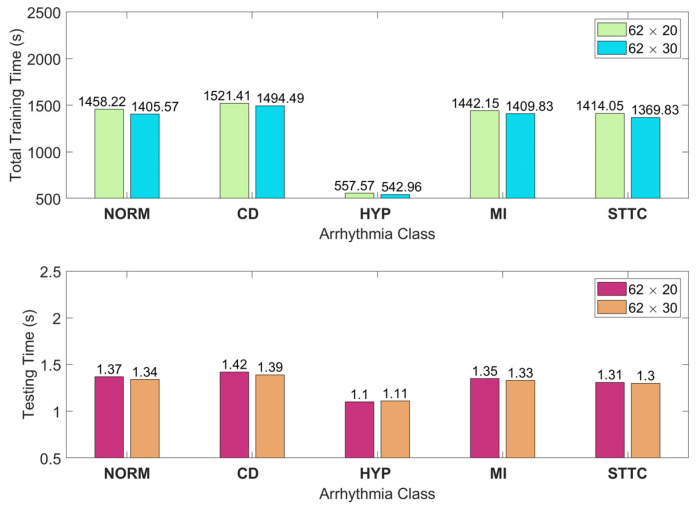
The training and testing times in a class-wise manner for the classification of five arrhythmias from the PTB-XL database. Here, testing times are per patient.

**Table 1 sensors-25-01220-t001:** A list of heartbeat types, labels, and numbers from the MIT-BIH and Chapman databases.

Database	Heartbeat Classes	Heartbeat Types	Labels	# of Samples
MIT-BIH	Non-ectopic beats (N)	Normal beats	N	89,836
		Left bundle branch block beats	L	
		Right bundle branch block beats	R	
		Nodal (junctional) escape beats	j	
		Atrial escape beats	e	
	Supraventricular ectopic beats (S)	Aberrated atrial premature beats	a	2781
		Supraventricular premature beats	S	
		Atrial premature contraction	A	
		Nodal (junctional) premature beats	J	
	Ventricular ectopic beats (V)	Ventricular flutter wave	!	7008
		Ventricular escape beats	E	
		Premature ventricular contraction	V	
	Fusion beats (F)	Fusion of ventricular and normal beats	F	802
	Unknown beats (Q)	Paced beats	/	15
		Unclassifiable beats	Q	
		Fusion of paced and normal beats	f	
Chapman	Sinus bradycardia (SB)	Sinus bradycardia	SB	3889
	General supraventricular tachycardia (GSVT)	Sinus tachycardia	ST	2307
		Supraventricular tachycardia	SVT	
		Atrial tachycardia	AT	
		Atrioventricular node re-entrant tachycardia	AVNRT	
		Atrioventricular re-entrant tachycardia	AVRT	
		Sinus atrium to atrial wandering rhythm	SAAWR	
	Atrial fibrillation (AFIB)	Atrial fibrillation beats	AFIB	2225
		Atrial flutter	AF	
	Sinus rhythm (SR)	Sinus rhythm beats	SR	2225
		Sinus irregularity beats	SI	

**Table 2 sensors-25-01220-t002:** The number of beats supplied from the MIT-BIH arrhythmia database under the inter-patient paradigm.

**Datasets**	**N**	**L**	**R**	**V**	**S**	**F**	**Data Records**
DS1 (Training set)	1000	1000	1000	1000	944	414	101, 106, 108, 109, 112, 114, 115, 116, 118, 119, 122, 124, 201, 203, 205, 207, 208, 209, 215, 220, 223, 230
DS2 (Test set)	36,426	4124	3475	3220	1837	388	100, 103, 105, 111, 113, 117, 121, 123, 200, 202, 210, 212, 213, 214, 219, 221, 222, 228, 231, 232, 233, 234

**Table 3 sensors-25-01220-t003:** The simulation results for the single and multiple runs of the proposed technique conducted on the MIT-BIH database under the inter-patient paradigm. The values in brackets are the mean values with standard deviations that characterize the multiple runs.

Size	Class	Sen (%)	Spe (%)	Ppr (%)	F1-Score (%)
62 × 20	N	99.98 (99.95 ± 2.65 × 10^−2^)	100.00 (99.85 ± 3.53 × 10^−1^)	100.00 (99.95 ± 1.26 × 10^−1^)	99.99 (99.95 ± 6.45 × 10^−2^)
	L	99.81 (99.28 ± 1.14 × 10^0^)	99.97 (100.00 ± 3.14 × 10^−3^)	99.69 (99.99 ± 3.45 × 10^−2^)	99.75 (99.63 ± 5.78 × 10^−1^)
	R	99.97 (99.98 ± 1.36 × 10^−2^)	99.94 (99.93 ± 5.39 × 10^−2^)	99.23 (99.12 ± 6.91 × 10^−1^)	99.60 (99.55 ± 3.51 × 10^−1^)
	V	99.94 (99.96 ± 2.63 × 10^−2^)	99.99 (99.97 ± 5.93 × 10^−2^)	99.88 (99.55 ± 8.17 × 10^−1^)	99.91 (99.75 ± 4.17 × 10^−1^)
	S	99.95 (99.98 ± 2.98 × 10^−2^)	99.87 (99.86 ± 3.86 × 10^−2^)	96.68 (96.60 ± 9.35 × 10^−1^)	98.29 (98.26 ± 4.83 × 10^−1^)
	F	77.58 (79.65 ± 4.48 × 10^0^)	100.00 (100.0 ± 0.00)	100.00 (100.0 ± 0.00)	87.37 (88.60 ± 3.03 × 10^0^)
62 × 30	N	99.95 (99.92 ± 5.94 × 10^−2^)	99.79 (99.96 ± 8.95 × 10^−2^)	99.92 (99.99 ± 3.07 × 10^−2^)	99.94 (99.95 ± 2.92 × 10^−2^)
	L	99.98 (99.95 ± 1.46 × 10^−1^)	100.00 (100.00 ± 2.21 × 10^−3^)	100.00 (99.98 ± 2.43 × 10^−2^)	99.99 (99.97 ± 7.64 × 10^−2^)
	R	100.00 (99.93 ± 1.27 × 10^−1^)	99.98 (99.99 ± 5.85 × 10^−3^)	99.68 (99.91 ± 7.73 × 10^−2^)	99.84 (99.92 ± 6.95 × 10^−2^)
	V	100.00 (100.00 ± 0.00)	99.94 (99.84 ± 6.62 × 10^−2^)	99.11 (97.72 ± 9.03 × 10^−1^)	99.55 (98.85 ± 4.63 × 10^−1^)
	S	100.00 (99.99 ± 2.33 × 10^−2^)	99.98 (99.98 ± 1.11 × 10^−2^)	99.51 (99.48 ± 2.82 × 10^−1^)	99.76 (99.73 ± 1.42 × 10^−1^)
	F	85.05 (84.42 ± 4.44 × 10^0^)	100.00 (100.00 ± 0.00)	100.00 (100.00 ± 0.00)	91.92 (91.49 ± 2.67 × 10^0^)

**Table 4 sensors-25-01220-t004:** A comparison of the proposed method with the up-to-date ECG arrhythmia classification methods that use the MIT-BIH database under the inter-patient paradigm. All metrics are given in the unit of percentage (%).

Ref.	# of Samples Train/Test	# of Classes	Class	Sen	Spe	Ppr	F1-Score	OA
[57]	51,013/41,472	5	V	97.60	99.80	97.60	97.60	98.80
			S	81.50	99.80	94.90	87.70	
[58]	51,013/49,705	4	V	93.45	^†^ 99.56	93.59	93.52	96.15
			S	76.21	^†^ 98.14	62.70	68.80	
[59]	50,928/49,646	5	V	73.92	^†^ 97.98	71.67	^†^ 72.78	^†^ 94.16
			S	70.26	99.44	82.90	^†^ 76.06	
[60]	66,354/49,661	4	V	^†^ 93.32	^†^ 99.46	^†^ 92.26	^†^ 92.79	95.60
			S	^†^ 89.32	^†^ 97.05	^†^ 53.77	^†^ 67.13	
[61]	50,969/49,661	4	V	72.26	^†^ 99.69	94.09	81.74	^†^ 92.81
			S	27.12	^†^ 97.83	32.44	29.55	
[62]	13,719/7942	5	V	88.35	^†^ 94.93	^†^ 79.82	^†^ 83.87	^†^ 89.00
			S	35.22	98.83	65.88	^†^ 45.94	
[63]	50,815/49,507	4	V	91.43	^†^ 99.41	91.46	^†^ 91.44	96.60
			S	85.64	^†^ 98.20	64.58	^†^ 73.63	
[64]	50,999/49,690	5	V	^†^ 91.06	^†^ 99.14	^†^ 88.03	^†^ 89.52	^†^ 98.01
			S	^†^ 83.44	^†^ 99.86	^†^ 95.69	^†^ 89.15	
[65]	69,876/30,713	4	V	95.96	99.68	94.54	95.24	^†^ 98.86
			S	92.94	99.63	87.31	90.04	
[66]	50,943/49,600	5	V	99.97	99.96	99.38	^†^ 99.67	^†^ 99.62
			S	99.56	99.68	92.23	^†^ 95.75	
[67]	110,844/49,661	4	V	93.46	^†^ 99.31	90.04	91.72	^†^ 97.50
			S	83.06	^†^ 99.32	82.48	82.77	
Proposed (62 × 20)	5358/49,470	6	V	99.96	99.97	99.55	99.75	99.73
			S	99.98	99.86	96.60	98.26	
Proposed (62 × 30)	5358/49,470	6	V	100.00	99.84	97.72	98.85	99.81
			S	99.99	99.98	99.48	99.73	

^†^ The results are calculated from the confusion matrix given in the related study.

**Table 5 sensors-25-01220-t005:** The simulation results for the single and multiple runs of the proposed technique using the raw ECG recordings of the Chapman database. The values in brackets are the mean values with standard deviations that characterize the multiple runs.

Size	Class	Sen (%)	Spe (%)	Ppr (%)	F1-Score (%)
62 × 20	SB	100.00 (100.00 ± 0.00)	100.00 (100.00 ± 0.00)	100.00 (100.00 ± 0.00)	100.00 (100.00 ± 0.00)
	GSVT	99.69 (99.26 ± 2.37 × 10^−1^)	100.00 (100.00 ± 0.00)	100.00 (100.00 ± 0.00)	99.85 (99.63 ± 1.19 × 10^−1^)
	AFIB	99.92 (99.92 ± 2.34 × 10^−14^)	99.80 (99.76 ± 3.37 × 10^−2^)	99.11 (98.95 ± 1.46 × 10^−1^)	99.51 (99.43 ± 7.37 × 10^−2^)
	SR	99.43 (99.56 ± 1.17 × 10^−1^)	99.98 (99.94 ± 2.37 × 10^−2^)	99.92 (99.75 ± 1.05 × 10^−1^)	99.67 (99.65 ± 7.53 × 10^−2^)
62 × 30	SB	99.69 (99.66 ± 5.89 × 10^−2^)	100.00 (99.98 ± 1.37 × 10^−2^)	100.00 (99.98 ± 1.79 × 10^−2^)	99.84 (99.82 ± 3.28 × 10^−2^)
	GSVT	84.85 (85.13 ± 2.68 × 10^−1^)	100.00 (100.00 ± 0.00)	100.00 (100.00 ± 0.00)	91.80 (91.97 ± 1.56 × 10^−1^)
	AFIB	100.00 (100.00 ± 0.00)	95.24 (95.19 ± 6.42 × 10^−2^)	82.60 (82.44 ± 1.93 × 10^−1^)	90.47 (90.38 ± 1.16 × 10^−1^)
	SR	95.43 (94.87 ± 4.38 × 10^−1^)	99.91 (99.90 ± 2.33 × 10^−2^)	99.57 (99.54 ± 1.08 × 10^−1^)	97.46 (97.14 ± 2.26 × 10^−1^)

**Table 6 sensors-25-01220-t006:** The simulation results for the single and multiple runs of the proposed technique using the de-noised ECG recordings of the Chapman database. The values in brackets are the mean values with standard deviations that characterize the multiple runs.

Size	Class	Sen (%)	Spe (%)	Ppr (%)	F1-Score (%)
62 × 20	SB	42.96 (43.15 ± 7.18 × 10^−1^)	99.97 (99.98 ± 1.30 × 10^−2^)	99.92 (99.94 ± 3.86 × 10^−1^)	60.08 (60.27 ± 6.99 × 10^−1^)
	GSVT	99.53 (99.64 ± 9.98 × 10^−2^)	63.55 (63.64 ± 4.90 × 10^−1^)	39.32 (39.41 ± 3.30 × 10^−1^)	56.37 (56.48 ± 3.44 × 10^−1^)
	AFIB	90.94 (90.94 ± 0.00)	100.00 (100.00 ± 0.00)	100.00 (100.00 ± 0.00)	95.25 (95.25 ± 0.00)
	SR	81.71 (82.28 ± 9.95 × 10^−1^)	99.22 (99.38 ± 7.01 × 10^−2^)	95.97 (96.80 ± 3.64 × 10^−1^)	88.27 (88.95 ± 6.55 × 10^−1^)
62 × 30	SB	99.79 (99.75 ± 4.26 × 10^−2^)	99.97 (99.97 ± 1.79 × 10^−2^)	99.97 (99.97 ± 2.31 × 10^−2^)	99.88 (99.86 ± 1.37 × 10^−2^)
	GSVT	98.66 (97.84 ± 4.61 × 10^−1^)	100.00 (99.99 ± 9.67 × 10^−3^)	100.00 (99.95 ± 4.17 × 10^−2^)	99.32 (98.88 ± 2.43 × 10^−1^)
	AFIB	100.00 (100.00 ± 0.00)	99.55 (99.35 ± 1.18 × 10^−1^)	98.08 (97.24 ± 4.91 × 10^−1^)	99.03 (98.60 ± 2.52 × 10^−1^)
	SR	99.84 (99.84 ± 9.43 × 10^−2^)	100.00 (100.00 ± 7.84 × 10^−3^)	100.00 (99.98 ± 3.45 × 10^−2^)	99.92 (99.91 ± 5.71 × 10^−2^)

**Table 7 sensors-25-01220-t007:** A comparison of the proposed method with the up-to-date ECG arrhythmia classification methods that use the Chapman database. All metrics are given in the unit of percentage (%).

Ref.	# of Samples Train/Test	Classes	Sen	Spe	Ppr	F1-Score	**OA**
[68]	9529/1059	SB	98.75	98.93	98.25	98.50	^†^ 96.13
		GSVT	89.94	99.30	96.75	93.22	
		AFIB	96.17	98.30	94.16	95.15	
		SR	96.88	98.32	93.96	95.40	
[69]	8516/2130	SB	97.81	97.93	96.45	97.13	^†^ 88.83
		GSVT	95.45	91.97	76.70	85.05	
		AFIB	69.21	96.62	84.38	76.05	
		SR	85.84	98.87	95.26	90.31	
[70]	7148/2043	SB	99.06	99.38	^†^ 98.93	^†^ 98.99	95.11
		GSVT	92.00	97.77	^†^ 92.58	^†^ 92.29	
		AFIB	92.08	97.00	^†^ 89.45	^†^ 90.75	
		SR	94.74	99.46	^†^ 97.56	^†^ 96.13	
^1^ Proposed	4000/6646	SB	100.00	100.00	100.00	100.00	99.76
		GSVT	99.26	100.00	100.00	99.63	
		AFIB	99.92	99.76	98.95	99.43	
		SR	99.56	99.94	99.75	99.65	
^2^ Proposed	4000/6606	SB	99.75	99.97	99.97	99.86	99.45
		GSVT	97.84	99.99	99.95	98.88	
		AFIB	100.00	99.35	97.24	98.60	
		SR	99.84	100.00	99.98	99.91	

^1^ The mean performance metrics satisfying the best statistics for the raw ECG records. ^2^ The mean performance metrics satisfying the best statistics for the de-noised ECG records. ^†^ The results are calculated from the confusion matrix in the related study.

**Table 8 sensors-25-01220-t008:** The simulation results for the single run and ten independent runs of the proposed technique using the ECG recordings of the PTB-XL database. The values in brackets are the mean values with standard deviations that characterize the multiple runs.

Size	Class	Sen (%)	Spe (%)	Ppr (%)	F1-Score (%)
62 × 20	NORM	99.52 (99.62 ± 1.26 × 10^−1^)	98.87 (99.50 ± 4.02 × 10^−1^)	99.48 (99.77 ± 1.86 × 10^−1^)	99.50 (99.69 ± 1.48 × 10^−1^)
	CD	97.46 (98.79 ± 7.97 × 10^−1^)	99.28 (99.22 ± 1.27 × 10^−1^)	89.61 (89.01 ± 1.60 × 10^0^)	93.37 (93.63 ± 9.27 × 10^−1^)
	HYP	0.00 (0.71 ± 9.92 × 10^−1^)	100.00 (100.00 ± 0.00)	NaN	NaN
	MI	99.28 (99.18 ± 1.77 × 10^−1^)	99.75 (99.73 ± 1.36 × 10^−1^)	98.32 (98.23 ± 8.82 × 10^−1^)	98.80 (98.70 ± 5.25 × 10^−1^)
	STTC	98.43 (98.86 ± 2.82 × 10^−1^)	99.74 (99.81 ± 8.23 × 10^−2^)	98.08 (98.61 ± 6.02 × 10^−1^)	98.25 (98.73 ± 3.89 × 10^−1^)
62 × 30	NORM	75.57 (76.40 ± 7.02 × 10^0^)	98.58 (99.01 ± 4.09 × 10^−1^)	99.14 (99.39 ± 2.57 × 10^−1^)	85.77 (86.25 ± 4.43 × 10^0^)
	CD	98.87 (99.21 ± 4.22 × 10^−1^)	76.26 (77.14 ± 5.01 × 10^0^)	21.01 (22.35 ± 4.36 × 10^0^)	34.65 (36.31 ± 5.59 × 10^0^)
	HYP	0.00 (0.82 ± 9.69 × 10^−1^)	100.00 (100.00 ± 0.00)	NaN	NaN
	MI	71.87 (72.77 ± 4.40 × 10^0^)	99.08 (98.75 ± 7.52 × 10^−1^)	92.13 (90.00 ± 5.34 × 10^0^)	80.75 (80.32 ± 3.05 × 10^0^)
	STTC	78.21 (76.53 ± 6.33 × 10^0^)	98.80 (99.62 ± 1.26 × 10^−1^)	98.12 (96.42 ± 1.15 × 10^0^)	87.04 (85.20 ± 4.22 × 10^0^)

**Table 9 sensors-25-01220-t009:** A performance comparison of the proposed FDM-CMA method with other up-to-date methods, in terms of sensitivity (Sen), F1-score, and overall accuracy, in classifying multi-type arrhythmias from the MIT-BIH, Chapman, and PTB-XL databases.

Methods	Year	Database	# of Samples **Train/Test**	Sen (%)	F1-Score (%)	OA (%)
Xu et al. [58]	2023	MIT-BIH	51,013/49,705	67.56	66.24	96.15
Zhou et al. [60]	2024		66,354/49,661	NaN	NaN	95.60
Xia et al. [61]	2023		50,969/49,661	50.97	53.53	92.81
Liu et al. [63]	2024		50,815/49,507	85.06	83.39	96.60
Jangra et al. [65]	2023		69,876/30,713	91.06	92.05	98.86
Xu et al. [67]	2022		110,844/49,661	78.32	82.02	97.46
Proposed *	-		5358/49,470	97.37	98.32	99.81
Yidirim et al. [68]	2020	Chapman	9529/1059	95.44	95.57	96.13
Lee et al. [69]	2024		8516/2130	87.08	87.13	88.83
Shi et al. [70]	2024		7148/2043	94.47	94.54	95.11
Proposed *	-	Chapman (Raw)	4000/6646	99.69	99.68	99.76
		Chapman (De-noised)	4000/6606	99.36	99.31	99.45
Shi et al. [70]	2024	PTB-XL	12,978/1652	57.15	58.60	77.06
Proposed *	-		4450/11,794	79.43	NaN	98.71

* The mean performance metrics satisfying the best statistics.

**Table 10 sensors-25-01220-t010:** The performance of the proposed technique using different feature matrix configurations corresponding to $D_{1}$, $D_{2}$, $D_{3}$, and D data matrices created using the ECG signals obtained from the MIT-BIH, Chapman, and PTB-XL databases. The metrics—sensitivity (Sen), F1-score, and overall accuracy (OA)—used for performance comparisons are given in the unit of percentage.

Data Matrix	*d*	MIT-BIH			Chapman—Denoised			Chapman—Raw			PTB-XL		
		OA	Sen	F1-Score	OA	Sen	F1-Score	OA	Sen	F1-Score	OA	Sen	F1-Score
**D** _1_	10	0.78	16.67	NaN	37.54	42.76	36.01	30.17	38.05	27.64	0.72	20.00	NaN
	20	92.91	73.86	NaN	42.81	50.08	44.51	84.50	84.13	82.88	59.62	48.37	NaN
	30	91.41	76.10	71.80	62.97	68.19	64.56	85.27	84.29	83.47	39.11	40.64	NaN
	40	91.01	79.62	74.76	85.48	84.84	83.94	87.47	85.99	85.34	35.82	39.72	32.01
**D** _2_	10	0.78	16.67	NaN	27.34	33.09	27.34	29.57	36.91	25.80	0.72	20.00	NaN
	20	97.77	82.11	NaN	40.98	47.73	42.21	85.81	85.26	84.14	59.85	48.23	NaN
	30	98.60	83.49	83.44	75.70	78.72	75.33	84.23	83.05	82.16	39.37	40.68	NaN
	40	95.85	83.22	81.80	88.62	87.73	86.97	87.68	86.42	85.71	38.43	40.49	32.85
**D** _3_	10	0.78	16.67	NaN	28.47	30.85	28.13	24.36	30.53	22.60	0.72	20.00	NaN
	20	99.18	89.38	91.51	52.42	62.40	58.33	99.61	99.60	99.54	98.56	79.09	NaN
	30	99.64	94.65	96.39	98.15	98.59	98.11	96.10	95.28	95.07	70.88	63.27	55.57
	40	99.49	94.28	95.98	95.79	95.32	95.06	93.09	91.89	91.53	46.39	46.94	40.06
**D**	10	0.78	16.67	NaN	27.23	32.54	25.81	29.55	37.13	27.13	0.72	20.00	NaN
	20	99.73	96.80	97.84	60.66	68.25	66.11	99.71	99.73	99.67	98.72	79.36	NaN
	30	99.70	97.32	98.16	99.49	99.56	99.44	96.42	95.72	95.55	74.37	65.01	NaN
	40	99.71	96.96	97.96	97.44	97.04	96.96	93.74	92.72	92.40	52.15	50.43	43.37

## Data Availability

All datasets used in this study was fetched from publicly available PhysioNet databases.

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
