# Peer review of "Automated ECG Arrhythmia Classification Using Feature Images with Common Matrix Approach-Based Classifier"

_sensors, 2025, doi:10.3390/s25041220_

Round 1

Reviewer 1 Report

Comments and Suggestions for Authors

1. When elaborating on the research innovation points, further emphasize the essential differences in the core mechanism from existing methods. For instance, detail the unique advantages of FDM over other time-frequency analysis methods (such as STFT, CWT, etc.) in extracting ECG signal features, not only in the aspect of no parameter setting, but also from the perspectives of adaptability to arrhythmia signals and feature expression capabilities. Strengthen this in the introduction and methods sections. Besides, more related work should be cited and discussed, such as Multi-scale model of effects of roughness on the cohesive strength of self-assembled monolayers, Unsupervised person re-identification based on adaptive information supplementation and foreground enhancement.
2. Explore the applicability of FDM to ECG data obtained from different types of arrhythmia signals (such as more complex arrhythmia combinations or rare arrhythmia types) and different acquisition devices. Supplement some preliminary experiments or theoretical analyses to demonstrate the method’s wide applicability, and mention this in the experimental section or the discussion of method limitations.
3. Conduct a more in-depth analysis of the computational complexity of CMA as a classifier, and compare it with other commonly used classifiers (such as SVM, CNN, etc.) in terms of time and space complexity. This will help readers better understand the efficiency advantages and potential bottlenecks of the method in practical applications, and add this content in the method introduction or experimental results and discussion sections.
4. When constructing the data matrix, it is recommended to explain in detail why the normalized zero-mean ECG signal and the absolute value of its FFT are chosen as specific rows for concatenation. Provide more theoretical basis or experimental verification to illustrate how this combination helps improve classification performance.
5. In addition to the study of data matrix configuration in the ablation experiment, conduct a sensitivity analysis on other key parameters in FDM and CMA that may affect model performance (such as the projection dimension d in 2DPCA, etc.). Show the trend of model performance changes under different parameter value ranges, providing readers with a more comprehensive understanding and application guidance of the model. Add relevant content in the experimental section.
6. When comparing with other methods, in addition to comparisons on each database separately, add an overall performance comparison and analysis across multiple databases. For example, calculate the comprehensive ranking of average accuracy, F1 value, and other indicators on all databases to more comprehensively highlight the advantages of this method. Set up a dedicated subsection in the results and discussion section for this analysis.
7. It is recommended to further explore the potential and possible challenges of this method in clinical practical applications. For instance, analyze its feasibility in real-time monitoring scenarios, including data processing speed and requirements for hardware devices; discuss the application prospects in different clinical environments (such as primary care units and large hospitals), and elaborate on this in the conclusion or future work section.
8. It is suggested to add methods for verifying the stability of experimental results, such as conducting multiple independent repeated experiments and using statistical tests (such as t-tests, analysis of variance, etc.) to verify the significant differences in results between different experiments. This will enhance the reliability and persuasiveness of the experimental results, and supplement relevant statistical analysis content in the experimental section.

Author Response

Manuscript ID:

Sensors-3446966

Title:

Automated ECG Arrhythmia Classification Using Feature Images with Common Matrix Approach-Based Classifier

Journal:

Sensors

We thank the reviewers for their insightful comments on the submitted manuscript. In light of reviewers’ comments, we revised the manuscript carefully. Revisions made in the manuscript are highlighted in red color. Replies to the reviewers are listed as follows:

REVIEWER #1:

Comment 1.1

When elaborating on the research innovation points, further emphasize the essential differences in the core mechanism from existing methods. For instance, detail the unique advantages of FDM over other time-frequency analysis methods (such as STFT, CWT, etc.) in extracting ECG signal features, not only in the aspect of no parameter setting, but also from the perspectives of adaptability to arrhythmia signals and feature expression capabilities. Strengthen this in the introduction and methods sections. Besides, more related work should be cited and discussed, such as Multi-scale model of effects of roughness on the cohesive strength of self-assembled monolayers, Unsupervised person re-identification based on adaptive information supplementation and foreground enhancement.

Reply 1.1

We thank the reviewer’s valuable comment. Taking this comment into consideration, in the introduction section (in page 4, at lines between 145 and 160), the following revision was made:

“This method decomposes a given signal into a finite number of band-limited orthogonal components, termed analytic Fourier intrinsic band functions (AFIBFs) [35]. Unlike the STFT and CWT, the FDM provides high-resolution T-F representations reflecting the characteristics of the considered signal through the AFIBFs [35–38]. The T-F representation of the signal is constructed using the instantaneous amplitudes (IAs) and instantaneous phases (IPs) of these complex-valued AFIBFs. Each one of the AFIBs represents a different frequency band and contains distinctive information that characterizes the signal. Taking this fact into consideration, the FDM has been applied to the ECG arrhythmia classification task. The ECG signals representing each arrhythmia class have unique characteristics and are defined by different numbers of distinct AFIBFs. The T-F representations created using these AFIBFs are decisive for the definition of arrhythmia classes. The ability to identify a signal via AFIBFs makes the FDM attractive for classification applications, such as in recognizing hand movements [39] and in detecting epileptic seizures [40], alcoholism [41], myocardial infarction [42], sleep apnea [43, 44], biometric identity [45], and hypertension [46].”

The reviewer recommended the following references for citing:

-             C. Zhang, A. P. Awasthi, J. Sung, P. H. Geubelle, and N. R. Sottos, “Multi-scale model of effects of roughness on the cohesive strength of self-assembled monolayers,” International Journal of Fracture, vol. 208, no. 1-2, pp. 131-143, Dec. 2017.

Briefly, this manuscript is related to the investigation of the influence of surface roughness on cohesive strength of an interface between a silica/self-assembled monolayer (Si/SAM) substrate and a transfer-printed gold film.

-             Q. Wang, Z. H. Huang, H. J. Fan, S. P. Fu, and Y. D. Tang, “Unsupervised person re-identification based on adaptive information supplementation and foreground enhancement,” IET Image Processing, vol. 18, no. 14, pp. 4680-4694, Dec. 2024.

Briefly, this manuscript is related to unsupervised person re-identification task.

We thank the valuable referee for these suggestions, but these references are not relevant to our proposed method and problem topic.

Comment 1.2

Explore the applicability of FDM to ECG data obtained from different types of arrhythmia signals (such as more complex arrhythmia combinations or rare arrhythmia types) and different acquisition devices. Supplement some preliminary experiments or theoretical analyses to demonstrate the method’s wide applicability, and mention this in the experimental section or the discussion of method limitations.

Reply 1.2

Three public databases—MIT-BIH, Chapman, and PTB-XL—were used to assess the performance of the proposed technique and, consequently, the FDM. These databases are different from each other, as their publishers declare. Different ECG recordings describing various arrhythmia types can be found in each of these databases. In the current paper, extensive analysis of different types of ECG signals provided from these three databases has already been presented to demonstrate the effectiveness of the proposed method.

As mentioned in Introduction section, the FDM is a data-driven adaptive signal decomposition tool that can be applied to non-linear and non-stationary time series and has been used to provide the T-F representations of ECG signals. It is free of parameter settings and has a precise mathematical basis. This method decomposes a given signal into a finite number of band-limited orthogonal components, termed analytic Fourier intrinsic band functions (AFIBFs) [35].

Comment 1.3

Conduct a more in-depth analysis of the computational complexity of CMA as a classifier, and compare it with other commonly used classifiers (such as SVM, CNN, etc.) in terms of time and space complexity. This will help readers better understand the efficiency advantages and potential bottlenecks of the method in practical applications, and add this content in the method introduction or experimental results and discussion sections.

Reply 1.3

The time consumptions required by the proposed FDM-CMA technique for classifying different types of arrhythmia from three public databases are given in Results and Discussion section for each conducted experiment. Associated with this issue, Figures 6, 9 and 11 are included to the revised paper.

Comment 1.4

When constructing the data matrix, it is recommended to explain in detail why the normalized zero-mean ECG signal and the absolute value of its FFT are chosen as specific rows for concatenation. Provide more theoretical basis or experimental verification to illustrate how this combination helps improve classification performance.

Reply 1.4

We thank the reviewer’s valuable comment. Taking this comment into consideration, in the Results and Discussion section, under Ablation Experiments subsection presented in pages 25 and 26, Ablation experiments were conducted to examine the impact of various data matrix configurations on the classification performance of the proposed FDM-CMA technique.

Comment 1.5

In addition to the study of data matrix configuration in the ablation experiment, conduct a sensitivity analysis on other key parameters in FDM and CMA that may affect model performance (such as the projection dimension d in 2DPCA, etc.). Show the trend of model performance changes under different parameter value ranges, providing readers with a more comprehensive understanding and application guidance of the model. Add relevant content in the experimental section.

Reply 1.5

We thank the reviewer’s valuable comment. Taking this comment into consideration, the impact of projection dimension demonstrates in Ablation Experiments subsection (in page 26) at Table 9.

Comment 1.6

When comparing with other methods, in addition to comparisons on each database separately, add an overall performance comparison and analysis across multiple databases. For example, calculate the comprehensive ranking of average accuracy, F1 value, and other indicators on all databases to more comprehensively highlight the advantages of this method. Set up a dedicated subsection in the results and discussion section for this analysis.

Reply 1.6

We thank the reviewer’s valuable comment. Taking this comment into consideration, we included a new subsection 4.4 titled “General Comparison with other recent studies” in pages 24 and 25.

Comment 1.7

It is recommended to further explore the potential and possible challenges of this method in clinical practical applications. For instance, analyze its feasibility in real-time monitoring scenarios, including data processing speed and requirements for hardware devices; discuss the application prospects in different clinical environments (such as primary care units and large hospitals), and elaborate on this in the conclusion or future work section.

Reply 1.7

There are some limitations in applying the proposed method to ECG signals. One of the main limitations of the FDM-CMA method is the imbalanced distribution of samples across classes in the training set. It requires ECG beat labels for classification task. The computational complexity of obtaining the T-F representations based on the FDM is dependent on the signal to be analyzed. The FDM spends a long time analyzing distorted signals, which restricts its practical applicability. This limitation can be overcome by exposing the signal to some pre-processing operations.

In the future, our research subjects will have two essential goals for practical application: employing tensor algebra to perform CMA and handling the processing load of the proposed FDM-CMA technique on cloud servers.

Comment 1.8

It is suggested to add methods for verifying the stability of experimental results, such as conducting multiple independent repeated experiments and using statistical tests (such as t-tests, analysis of variance, etc.) to verify the significant differences in results between different experiments. This will enhance the reliability and persuasiveness of the experimental results, and supplement relevant statistical analysis content in the experimental section.

Reply 1.8

We thank the reviewer’s valuable comment. Taking this comment into consideration, the performance of the proposed arrhythmia classification technique on each dataset was evaluated through the Monte Carlo cross-validation (MCCV). Mean performance metrics together with standard deviations were calculated by taking average of MCCV simulations performed on ECG recordings from three databases. In that vein, additional test results presented in Tables 5, 6, and 8 were included to the revised paper.

Reviewer 2 Report

Comments and Suggestions for Authors

This manuscript, "Automated ECG Arrhythmia Classification Using Feature Images with Common Matrix Approach Based Classifier" contains several significant issues that require careful revision.

The manuscript fails to provide a clear differentiation of its contributions from existing literature. While the authors claim their method outperforms recent works, the manuscript does not adequately justify how the proposed approach is unique compared to other techniques in terms of methodology or application.

Although the authors mention using Fourier decomposition method, two-dimensional principal component analysis, and creation of data matrices, manuscript lacks clarity regarding the specifics of these techniques. For instance:

1) How is Fourier decomposition method implemented and why was it chosen over other time-frequency methods?

2) What parameters were optimized in two-dimensional principal component analysis process?

3) Details about the common matrix approach and its integration with other components are vague.

This lack of specificity makes it challenging to assess the technical rigor of the work.

The manuscript focuses heavily on reporting high accuracy rates (e.g., 99.82% with MIT-BIH, 99.59% with Chapman) but does not provide adequate context about the experimental setup:

1) Were the databases pre-processed uniformly?

2) Was the inter-patient paradigm rigorously maintained?

3) Were the reported metrics validated using cross-validation or an independent test set to prevent overfitting?

Without these critical clarifications, the validity of the results remains questionable.

The study is based solely on public datasets (MIT-BIH, Chapman, PTB-XL), which, while valuable, limits its real-world applicability. The manuscript does not explore whether the method can generalize to diverse patient populations, device settings, or real-time clinical scenarios.

The manuscript fails to highlight how this approach can address practical challenges in clinical environments. For instance, how does the proposed model handle noisy or incomplete ECG recordings, which are common in real-world settings? This gap makes it hard to gauge the method's translational value.

The authors do not explicitly state whether ethical approvals or permissions for dataset usage were obtained. Given the clinical nature of the databases, this omission is significant and raises concerns about compliance with data-sharing protocols.

The manuscript does not address the interpretability of the proposed classifier. Models for medical decision-making must ensure transparency to gain trust from clinicians. The lack of explanation about how predictions are made (e.g., feature importance) weakens the applicability of the approach.

The authors claim their method outperforms recent efforts across multiple databases and arrhythmia types. However, without providing comparative benchmarks or references to those "recent works," these statements appear exaggerated and unsubstantiated.

The high accuracy rates reported (up to 99.82%) raise concerns about overfitting, especially since no details are provided about regularization techniques, hyperparameter tuning, or prevention of data leakage.

The manuscript emphasizes overall accuracy without addressing other critical performance metrics like sensitivity, specificity, or precision-recall trade-offs. These are crucial in medical diagnostics where false positives and false negatives carry significant implications.

Comments on the Quality of English Language

The English could be improved to more clearly express the research.

Author Response

Manuscript ID:

Sensors-3446966

Title:

Automated ECG Arrhythmia Classification Using Feature Images with Common Matrix Approach-Based Classifier

Journal:

Sensors

We thank the reviewers for their insightful comments on the submitted manuscript. In light of reviewers’ comments, we revised the manuscript carefully. Revisions made in the manuscript are highlighted in red color. Replies to the reviewers are listed as follows:

REVIEWER #2:

Comment 2.1

The manuscript fails to provide a clear differentiation of its contributions from existing literature. While the authors claim their method outperforms recent works, the manuscript does not adequately justify how the proposed approach is unique compared to other techniques in terms of methodology or application.

Reply 2.1

We stated the novelty of our FDM-CMA in the Introduction section. Specifically, we emphasized how this approach differs from existing techniques in terms of feature extraction, training and testing stage of the classifier. For instance, the use of feature images derived from the FDM allows for a more robust and interpretable representation of ECG signals, which has not been extensively explored in prior works. CMA is a parameter free classification method that works well under insufficient data case rather than ampful data case. We also included a detailed comparison with state-of-the-art methods in the Results and Discussion section, highlighting the methodological and practical advantages of our approach.

To better justify the claimed performance improvements, we expanded the Experiments section to include additional comparative analyses. This will involve Monte-Carlo cross validation (MCCV) on each publicly available ECG dataset to demonstrate generalizability. Providing a comprehensive comparison of performance metrics (e.g., accuracy, sensitivity, specificity, F1-score) against recent methods. Conducting ablation study to isolate the contribution of each component of feature matrices. To enhance clarity, we updated some figures to illustrate the workflow of the proposed method. We also added a subsection titled “General Comparison with other recent studies” to show the advantages of the proposed method.

Comment 2.2

Although the authors mention using Fourier decomposition method, two dimensional principal component analysis, and creation of data matrices, manuscript lacks clarity regarding the specifics of these techniques. For instance:

1) How is Fourier decomposition method implemented and why was it chosen over other time-frequency methods?

2) What parameters were optimized in two-dimensional principal component analysis process?

3) Details about the common matrix approach and its integration with other components are vague.

This lack of specificity makes it challenging to assess the technical rigor of the work.

Reply 2.2

We acknowledge that the manuscript does not provide sufficient specifics regarding the implementation of two-dimensional principal component analysis (2-D PCA) and drawbacks of the FDM and the CMA. We thank the reviewer’s valuable comment. Taking this comment into consideration,

1-)

We provided a detailed explanation of how the FDM is implemented in the Subsection 2.1. We barely included mathematical formulations to clarify the FDM process. Specifically, we will describe how the ECG signals are transformed into time-frequency representation. We explained why the Fourier decomposition method was chosen over other time-frequency methods (e.g., wavelet transform or short-time Fourier transform). For instance, Fourier decomposition was selected due to its flexibility to apply on every kind of signal. FDM does not have cross-term products.

2-)

We provided a step-by-step description of how 2-D PCA is calculated and applied to the data matrices to obtain feature images, including the mathematical formulation of the covariance matrix, eigenvalue decomposition, and projection onto the principal component space, in Subsection 2.1.2 and 2.1.3.

We included a reference [50] on how the dimension parameter in 2-D PCA is selected to enhance the efficiency and accuracy of the classification process.

We expanded the Ablation experiments to include additional comparative analyses to show the optimized parameter during the 2-D PCA process, such as the number of principal components retained.

3-)

A clear definition of what the Common Matrix represents, what CMA requires, how it is constructed from the training input data and how it discriminates the test data is included in the CMA section 2.2. We also provided schematic representations of applying CMA for training and testing.

Comment 2.3

The manuscript focuses heavily on reporting high accuracy rates (e.g., 99.82% with MIT-BIH, 99.59% with Chapman) but does not provide adequate context about the experimental setup:

1) Were the databases pre-processed uniformly?

2) Was the inter-patient paradigm rigorously maintained?

3) Were the reported metrics validated using cross-validation or an independent test set to prevent overfitting?

Without these critical clarifications, the validity of the results remains questionable.

Reply 2.3

Thank you for raising these critical points. We acknowledge the need for greater transparency in describing our experimental protocols and validation strategies. Below, we address each concern and outline revisions to ensure clarity and rigor in the revised manuscript:

1-)

  • MIT-BIH: Signals taken from this database were segmented to 301 sample as a pre-process. Because the records in this database are too long to process individually. No pre-processing other than segmentation is inherited with database.
  • Chapman: This database was released in the raw and de-noised formed by the owner. Lead-II signals were extracted out of the 12 Lead. Amplitude was scaled to [0 1]. Apart from this, no action as was taken a pre-processing.
  • PTB-XL: Lead-I signals were extracted out of the 12 Lead. Amplitude was scaled to [0 1]. Apart from this, no action was taken as a pre-processing.

2-)

Inter-Patient Splits: For All database, data was partitioned by patient, ensuring no patient’s samples appeared in both training and test sets.

3-)

We performed Monte Carlo cross-validation (100 random trials for MIT-BIH database, 10 random trials for Chapman and PTB-XL databases) with strict inter-patient splits to ensure no patient overlap between training and test sets. Performance metrics were averaged across multiple runs, with standard deviations reported to quantify stability. The model achieved consistent performance across all MCCV simulations demonstrating reliability against random data partitioning.

Comment 2.4

The study is based solely on public datasets (MIT-BIH, Chapman, PTB-XL), which, while valuable, limits its real-world applicability. The manuscript does not explore whether the method can generalize to diverse patient populations, device settings, or real-time clinical scenarios.

Reply 2.4

While public datasets like the MIT-BIH, Chapman and PTB-XL provide standardized benchmarks, they may not fully represent the variability encountered in clinical practice, such as heterogeneous patient demographics, ambulatory ECG artifacts, or device-specific signal characteristics. Future work will prioritize validation on multi-center, real-world datasets.

Comment 2.5

The manuscript fails to highlight how this approach can address practical challenges in clinical environments. For instance, how does the proposed model handle noisy or incomplete ECG recordings, which are common in real-world settings? This gap makes it hard to gauge the method's translational value.

Reply 2.5

The performance of the proposed method is also evaluated using noisy (raw) ECG recordings fetched from the Chapman database. It is demonstrated by MCCV simulations that the proposed FDM-CMA method yields satisfactory classification results. This issue is considered in the subsection 4.2.1.

Comment 2.6

The authors do not explicitly state whether ethical approvals or permissions for dataset usage were obtained. Given the clinical nature of the databases, this omission is significant and raises concerns about compliance with data-sharing protocols.

Reply 2.6

All datasets used in this study (MIT-BIH, Chapman, PTB-XL) are publicly available, de-identified, and explicitly permitted for academic research under their respective licenses.

  • MIT-BIH Arrhythmia Database:

o            Collected under Institutional Review Board (IRB) approval at MIT and Beth Israel Hospital, with data publicly shared under the PhysioNet Credentialed Health Data License (CHDL 1.5.0).

  • Chapman-Shaoxing Dataset:

o            Shared under the CC BY 4.0 license (Creative Commons Attribution) with no restrictions on non-commercial use.

o            Explicit permission granted by Zhejiang Provincial People’s Hospital Ethics Committee for open access.

  • PTB-XL:

o            Released under the Open Data Commons Attribution License (ODC-By), permitting redistribution and modification with proper attribution.

o            Ethics approval documented in the original publication  (Wagner et al. [54])

Comment 2.7

The manuscript does not address the interpretability of the proposed classifier. Models for medical decision-making must ensure transparency to gain trust from clinicians. The lack of explanation about how predictions are made (e.g., feature importance) weakens the applicability of the approach.

Reply 2.7

We have performed ablation studies to evaluate the contribution of individual components of feature matrices used by our classifier. By systematically removing or modifying parts of the feature matrices, we demonstrate how each component influences performance and interpretability. These results are presented in Subsection 4.5 and provide additional transparency into the model's inner workings.

Comment 2.8

The authors claim their method outperforms recent efforts across multiple databases and arrhythmia types. However, without providing comparative benchmarks or references to those "recent works," these statements appear exaggerated and unsubstantiated.

Reply 2.8

We have included detailed comparisons with state-of-the-art methods across multiple databases (e.g., MIT-BIH, Chapman, and PTB-XL) and arrhythmia types. Performance metrics such as overall accuracy, sensitivity and F1-score are now provided in Tables 4, 7, 9 along with the results of recent methods for direct comparison. To demonstrate the robustness of our results, we have performed MCCV simulations on each experiment.

Comment 2.9

The high accuracy rates reported (up to 99.82%) raise concerns about overfitting, especially since no details are provided about regularization techniques, hyperparameter tuning, or prevention of data leakage.

Reply 2.9

The proposed method do not have hyperparameter tuning. It is not possible to have data leakage because all experiments were performed considering independent MCCV simulations. Also it was worked taking the inter-patient paradigm into consideration. The reason for high accuracy rate is that the data augmentation which can increase the diversity of the training data reducing the risk of overfitting. Also we used the 2-D PCA to generate feature matrices from data matrices. It helps removing noise and redundant information which prevent overfitting and increasing discernibility between classes.

Comment 2.10

The manuscript emphasizes overall accuracy without addressing other critical performance metrics like sensitivity, specificity, or precision-recall trade-offs. These are crucial in medical diagnostics where false positives and false negatives carry significant implications.

Reply 2.10

Crucial performance metrics in medical diagnostics were included to the related comparison tables, as given in Tables 4, 7, 9.

Round 2

Reviewer 1 Report

Comments and Suggestions for Authors

The revised paper has better quality, and I agree to accept it.

Reviewer 2 Report

Comments and Suggestions for Authors

All my suggestions and recommendations are answered adequately.